# CHARACTERISTICS AND FREQUENCY OF LARGE SUBMARINE LANDSLIDES AT THE WESTERN TIP OF THE GULF OF CORINTH

Arnaud Beckers[1,2*], Aurelia Hubert-Ferrari[1], Christian Beck[2], George Papatheodorou[3], Marc de Batist[4], Dimitris Sakellariou[5], Efthymios Tripsanas[6], Alain Demoulin[1]

Department of Geography, University of Liège, allée du 6 août 2, 4000 Liège, Belgium. Email : beckersarnaud@gmail.com.
ISTerre, CNRS UMR 5275, University of Savoie, F-73376 Le Bourget du Lac, France.
Department of Geology, University of Patras, Greece
Department of Geology and Soil Science, Gent
Institute of Oceanography, Hellenic Center for Marine Research, GR-19013 Anavyssos, Greece
Gnosis Geosciences, Edinburgh, EH10 5JN, U.K.

* Now at: CSD Engineers, Namur Office Park 2, Avenue des dessus de Lives, 5101 Namur, Belgium

Correspondence to: Aurelia Hubert-Ferrari (aurelia.ferrari@ulg.ac.be)

## Abstract
Coastal and submarine landslides are frequent at the western tip of the Gulf of Corinth, where small to medium failure events ($10^6$-$10^7$ m$^3$) occur on average every 30-50 years. These landslides trigger tsunamis, and consequently represent a significant hazard. We use here a dense grid of high-resolution seismic profiles to realize an inventory of the large mass transport deposits (MTDs) that result from these submarine landslides. Six large mass wasting events are identified, and their associated deposits locally represent 30% of the sedimentation since 130ka in the main western Basin. In the case of a large MTD of ~1 km$^3$ volume, the simultaneous occurrence of different slope failures is inferred and suggests an earthquake triggering. However, the overall temporal distribution of MTDs would result from the time-dependent evolution of pre-conditioning factors, rather than from the recurrence of external triggers. Two likely main pre-conditioning factors are (1) the reloading time of slopes, which varied with the sedimentation rate, and (2) dramatic changes in water depth and water circulation that occurred 10-12ka ago during the last post-glacial transgression. Such sliding events likely generated large tsunami waves in the whole Gulf of Corinth, possibly larger than those reported in historical sources considering the observed volume of the MTDs.

## 1 Introduction
The study of marine geohazards through their imprint in the late Quaternary sedimentary record is of great significance, since it can provide further information on geohazard events recorded in historical records, or even extend this record to much earlier times. The identification and recurrence patterns of mass transport deposits (MTDs) resulting from submarine landslides in sedimentary basins and lakes provide valuable information on possibly associated tsunamis as well as their potential trigger (e.g. earthquake). Tsunami hazard is particularly an issue of concern in the Mediterranean Sea where more than 300 tsunamis have been listed in the historical and sedimentary records  (Soloviev, 1990; Salamon et al., 2007; Lorito et al., 2008).

This paper focuses on the Gulf of Corinth, Greece, located in the most seismically active part of the Corinth Rift. This area shows one of the largest seismic hazard in Europe (Woessner et al., 2013) and is affected by a tsunami once every 19 years on average, leading to a significant risk (Papadopoulos, 2003; Papathoma and Dominey-Howes, 2003). The gulf's western tip is the most active part of the Corinth rift, characterized by an extension of 15 mm.yr$^{-1}$ (Briole et al., 2000), and by frequent submarine or coastal landslides (e.g. Henzen et al., 1966; Papatheodorou and Ferentinos, 1997; Lykousis et al., 2009). Small to medium failure events ($10^6$-$10^7$ m$^3$) occur on average every 30-50 years (Lykousis et al., 2007a). These landslides trigger tsunamis (Galanopoulos et al., 1964; Stefatos et al., 2006; Tinti et al., 2007) and induce coastal erosion by upslope retrogression (Papatheodorou and Ferentinos, 1997, Hasiotis et al., 2006). Tsunamis reaching an intensity ≥ 4 consequently represent a significant hazard in the western Gulf of Corinth (Beckers et al. 2017), and are documented for the last two millennia from historical

sources and onland geological studies (De Martini et al., 2007; Kontopoulos and Avamidis, 2003;
Kortekaas et al., 2011). However, these data sets are incomplete.
A dense grid of high-resolution seismic profiles acquired in this area (Beckers et al., 2015) was used to
realize an inventory of the large mass transport deposits (MTDs) that may be interpreted as the result of
submarine landslides. Dated from the Late Pleistocene and the Holocene, the mapped mass transport
deposits range from $10^6$-$10^9$ m$^3$. Average recurrence intervals are presented and discussed, as well as
pre-conditioning factors that might have played a role in the occurrence of these large submarine
landslides. The MTDs' temporal distribution is discussed, as well as the implications of their occurrence
on tsunami hazard.
**2 Setting**
The western Gulf of Corinth is characterized by a relatively flat deep basin dipping gently to the east.
Featuring a narrow canyon in the west (the Mornos Canyon), it widens in the east (Delphic Plateau, Fig.
1). It is bordered by steep slopes on all sides (Fig. 1) To the north, it is limited by the Trizonia scarp with
slopes ranging from 25° to locally more than 35° and the associated Trizonia Fault (Nomikou et al.,
2011); these slopes are mostly devoid of sediments which are trapped in the bay areas to the north (Fig.
1B). To the south, the western Gulf is bordered by 400m high Gilbert deltas built by the Erineos,
Meganitis and Selinous rivers that lie in front of the active Psathopyrgos, Kamari and Aigion Faults
running along or near the coastline. Delta fronts have 15° to 35° slopes incised by gullies (Lykousis et
al., 2007; Nomikou et al, 2011) and consist of a thick pile of fine grained sediments. The delta-front
sediments accumulated over the Holocene and the previous glacial-interglacial period have thicknesses,
respectively, larger than 50m and 100 m (Fig. 1B and 1C; Beckers, 2015; Beckers et al, 2016). At the
north-western end of the Gulf, lies the largest fan-delta of the Mornos River that drains 913 km$^2$ and is
by far the largest watershed among the rivers flowing toward the westernmost Gulf of Corinth (Fig. 1A).
The delta fronts are highly unstable (Ferentinos et al, 1988; Lykousis et al., 2009), which favours
frequent submarine landsliding (Stefatos et al., 2006; Tinti et al., 2007; Fig. 1B). During the last
centuries, submarine landslides have been triggered by earthquakes and by sediment overloading on
steep slopes (Galanopoulos et al., 1964; Heezen et al., 1966). Numerous debris-flow deposits and mass-
transport deposits (MTDs) have thus accumulated at the foot of the deltas (Ferentinos et al., 1988;
Beckers et al., 2016; Fig. 1B). Alongside these gravity-driven sedimentary processes, contour-parallel
bottom-currents also influenced sediment transport in this area (Beckers et al., 2016).
The shallow sedimentary infill of Gulf of Corinth infill consists of a distinct alternation between
seismic-stratigraphic units with parallel, continuous high-amplitude reflections and units with parallel,
continuous low amplitude reflections to acoustically transparent seismic facies (e.g. Bell et al., 2008;
Taylor et al., 2011). Generally, the semi-transparent units are thicker than the highly reflective units (e.g.
Taylor et al., 2011). These alternating seismic-stratigraphic units have been observed throughout the
Gulf of Corinth and have been interpreted as depositional sequences linked to glacio-eustatic cycles
(Bell et al., 2008; Taylor et al., 2011). Because of the presence of the 62 m deep Rion Sill at the entrance
of the Gulf, the Gulf of Corinth was disconnected from the World Ocean during Quaternary lowstands
and was thus a non-marine sedimentary environment. The marine and non-marine environments are
associated with different climatic regimes (e.g. Leeder et al., 1998). During glacial stages, the sparse
vegetation cover was more favourable to erosion than during interglacials, so high quantities of
sediments were routed towards the Gulf (Collier et al., 2000). These lowstand deposits appear as thick,
low-reflective units. The thin, high-reflective units are interpreted to represent the marine highstand
deposits. The last lacustrine-marine transition has been sampled in different sedimentary cores (Collier
et al., 2000; Moretti et al., 2004; Van Welden, 2007; Campos et al., 2013).
**3 Data and Method**
Two seismic reflection surveys were carried out in 2011 and 2014 with the aim of imaging the
subsurface below the westernmost Gulf of Corinth floor. The data were acquired by the Renard Center
of Marine Geology of the University of Ghent along a grid of 600 km high-resolution seismic profiles
with a "CENTIPEDE" Sparker seismic source combined with a single-channel high-resolution streamer
as receiver (see details in Beckers et al., 2015, the seismic grid is shown in Fig. 2). The expected vertical
resolution at depth is ~1 m. In the deep basin (Canyon and Delphic Plateau areas, Fig. 1), the maximum
penetration depth below the sea floor is about 360 ms TWTT (two-way travel time) to the east and about
100 ms TWTT to the west, i.e., 270-360 m and 75-100 m, respectively.
The inferred stratigraphic framework (Beckers et al., 2015) permits to identify two temporal horizons.
Reflector 1 has been mapped in the whole study area, except in a basin west of the Trizonia Island (Fig.
1). This reflector corresponds to the beginning of the last post-glacial transgression, at 10.5-12.5 ka
(Cotterill, 2006; Beckers et al., 2016). The second temporal horizon, 'reflector 2', has been mapped in
the Delphic Plateau area only. It corresponds to the marine isotopic stage 6 to 5 transgression, which
occurred at ca. 130 ka.
**Figure 1.** Study area with at the top, the fault map of Beckers et al. (2015) with the bathymetry from
Nomikou et al. (2011), in the middle, the morphosedimentary map of Holocene deposits of Beckers et
al. (2016), at the bottom the isopach maps of the Holocene (Right ; Beckers et al., 2016), and of the
preceding period from 10 to 130 ka (Left ; Beckers, 2015). White areas in the bottom figures correspond
to the ones with poor data or with an absence of stratigraphic marker. Grey curves in middle and bottom
figures are sea floor contour lines interpolated from the seismic grid.

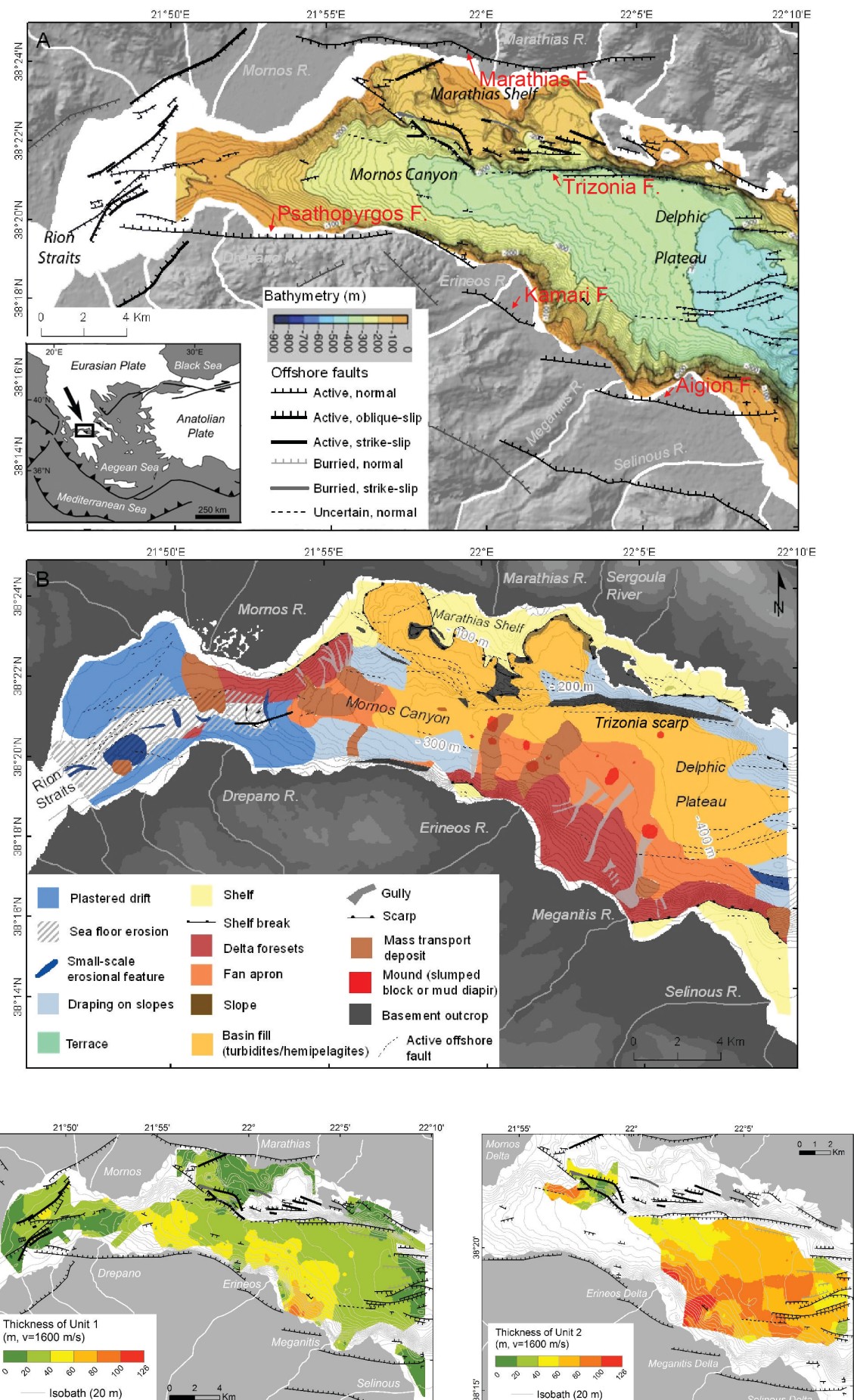


**Figure 2.** Inventory of mass transport deposits (MTDs) at the westernmost Gulf of Corinth for the last ca.
130 ka. A) spatial extent and age of the 32 MTDs with in grey seismic grid used for the inventory; B) to G):
spatial distribution of MTDs for each sliding event (SE). Grey lines show the seismic grid. Black dots
represents the mounds described in Beckers et al. (2016a). The total volume of sediments in the MTDs is

 mentioned for each sliding event.

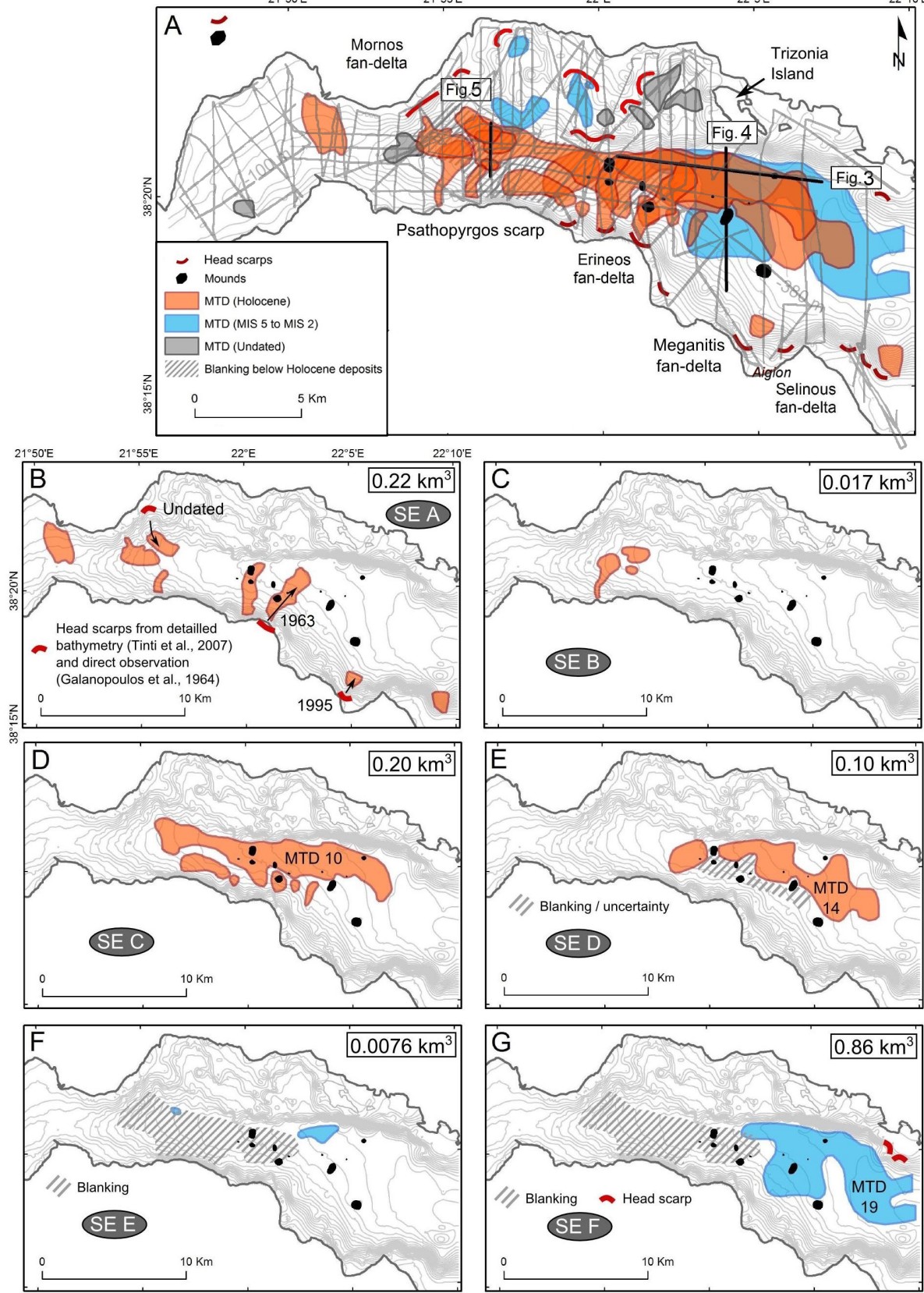

Mass transport deposits have been identified on high-resolution seismic profiles based on their typical
seismic facies made of discontinuous to chaotic reflections. The shape of each deposit in map view has
been interpolated manually, based on the seismic profiles that intersect the MTD. Thicknesses were
derived using a seismic velocity of 1600 m s$^{-1}$ (Bell et al., 2009). For the largest MTDs, an inverse
distance weighted interpolation between thickness data points was used to derive isopach maps of the
deposits and estimate their total volume. However, this interpolation method cannot be used for smaller
MTDs crossed only by a few seismic lines. In this case, the volume was estimated by multiplying the
MTD surface by an average thickness value. The derived volumes of small MTDs (surface area < ~2
km$^2$) are thus rough estimates, especially for MTDs crossed by only two or three seismic profiles. By
contrast, volume estimates of large MTDs (surface area > ~5 km$^2$) are more accurate with volume
uncertainties probably < 20 %.
Landslide headscarps have been mapped using three different data sources, namely (1) the grid of high-
resolution seismic profiles acquired for this study, (2) an analysis of three submarine landslides in the
study area by (Tinti et al., 2007), and (3) a 3D bathymetric view of the area between the Erineos and the
Selinous fan-deltas from Lykousis et al. (2009). In the absence of multi-beam bathymetry over the
whole study area, the mapping of Late Quaternary submarine landslides head scarps presented here is
certainly not exhaustive. The location of potential headscarps associated with the largest MTDs mapped
in the following are also discussed considering the location of the thickest deposits and the nearest
upslope delta-front sediments.
**4 Results**
Thirty-two MTDs have been imaged in the study area, from which 67% are located in the large E-W
trending basin located below the flat deep basin (Mornos Canyon and Delphic Plateau, Fig. 2). Eight
MTDs have been identified in the northern margin of the Gulf, and two in the Nafpaktos Bay to the west
of the Corinth Gulf (Fig. 2). The age of 24 MTDs has been estimated based on the stratigraphic
framework developed previously (Beckers et al., 2015): 19 of them occurred during the Holocene and 5
during the period between ~130 ka and ~11.5 ka. A finer stratigraphy could be established in the flat
deep basin, thanks to the relative continuity of the reflectors over this 20 km-wide area. Consequently,
this work focuses on the 22 MTDs located in this area.
In the Delphic Plateau basin (eastern part of the deep flat basin), most MTDs are imaged as lenticular
bodies of low-amplitude, incoherent reflections (Fig. 3 and 4). They generally have a flat upper surface
and pinch out on their margins. Their thickness ranges between a few meters, which is the minimal
thickness for a MTD to be imaged with the seismic system used, and 53 meters. The geometry and
seismic facies indicate subaquatic mass-flow deposits (e.g. Moernaut et al., 2011, Strasser et al., 2013).
The seismic facies of many MTDs also suggests a fine-grained lithology. However, this statement must
be viewed cautiously considering the uncertainties on the interpretation of seismic facies in terms of
grain-size, especially for reworked sediments. For instance, failure of coarse-grained deltaic deposits
commonly result to their total disaggregation and transformation into grain flows and turbidity currents,
whereas finer grained deposits evolve as landslides and cohesive debris flows (Tripsanas et al., 2008).
In the Mornos Canyon basin (western part of the deep flat basin), the MTDs present the same general
characteristics but the reflector pattern is more variable (Fig. 5). Some high-amplitude reflections and
coherent layering are observed in some MTDs, suggesting coarser-grained sediments and locally
preserved stratigraphy.
Finally, some of the 22 MTDs show sediment/fluid escape features at their top (Fig. 3 and 5). Such
features might have been produced by the combination of under-compaction (excess pore water
pressure) and shaking, thus possibly pointing to paleoearthquakes (e.g. Moernaut et al., 2007, Moernaut
et al., 2009). The volume of sediments in individual MTDs ranges from 7.7 10$^5$ to 8.6 10$^8$ m$^3$ (Fig. 6).
Landslide headscarps have been identified in different parts of the study area (Fig. 2A). They are
particularly numerous on the slopes of the large Gilbert fan-deltas of the Erineos, Meganitis and
Selinous at the south-east and Mornos at the north-west. In the latter area, one up to 50 m-high
headscarp is imaged in the seismic data. The absence of undisturbed sediments on the erosional slope,
downslope of the headscarp, suggests a recent age. In the Erineos, Meganitis and Selinous fan-delta
slopes, headscarps have been identified in the seismic data and on the 3D view from Lykousis et al.
(2009). Most of these headscarps are relatively small, lunate-shaped features linked to gullies (see also
the bathymetric map in Fig.1). Two large head scarps are localized on the northern slope as well (Fig.
2A). Linking a headscarp to a particular MTD is often delicate for two reasons. First, the age of the
headscarps is difficult to estimate because these erosional forms often affect steep slopes in coarse-
grained deposits, making impossible to define a seismic stratigraphy in such areas. Second, at the foot of
these erosional slopes, a high number of MTDs are stacked (e.g., Fig. 3). Exceptions, detailed hereafter,
concern three recent submarine landslides and the largest observed MTD (MTD 19 in sliding event F).

**Figure 3.** E-W Sparker seismic profile showing the mass transport deposits imaged in the Delphic Plateau
basin. See the location of the profile in Fig. 2. Horizon [1] indicates the beginning of the last post-glacial
transgression, at 10.5-12.5 ka and horizon [2] the marine isotopic stage 6 to 5 transgression, which
occurred at ca. 130 ka (Cotterill, 2006; Beckers et al., 2015; 2016)

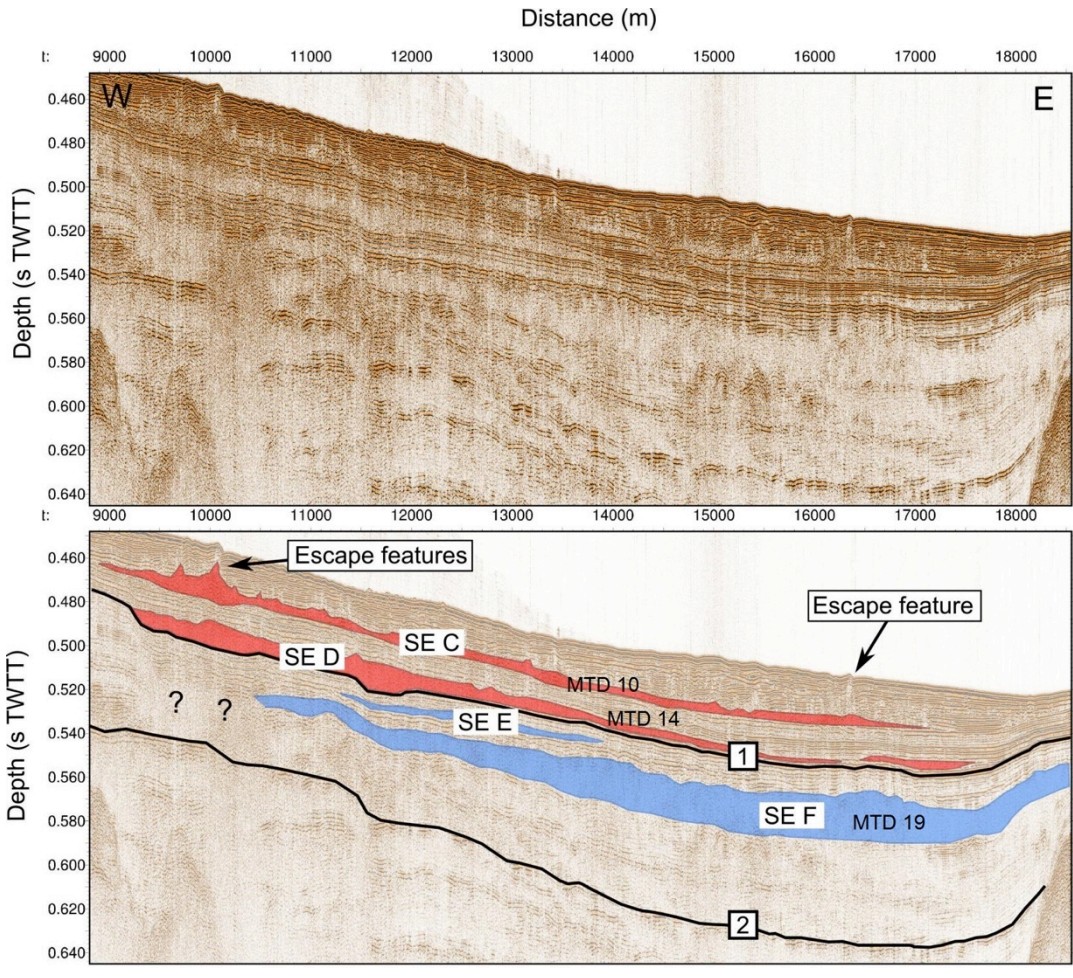


**Figure 4.** S-N Sparker seismic profile showing the mass transport deposits imaged in the Delphic Plateau
basin. Questions marks highlight units of remobilized sediments that are difficult to localize in the
stratigraphic framework. See the location of the profile in Fig. 2.

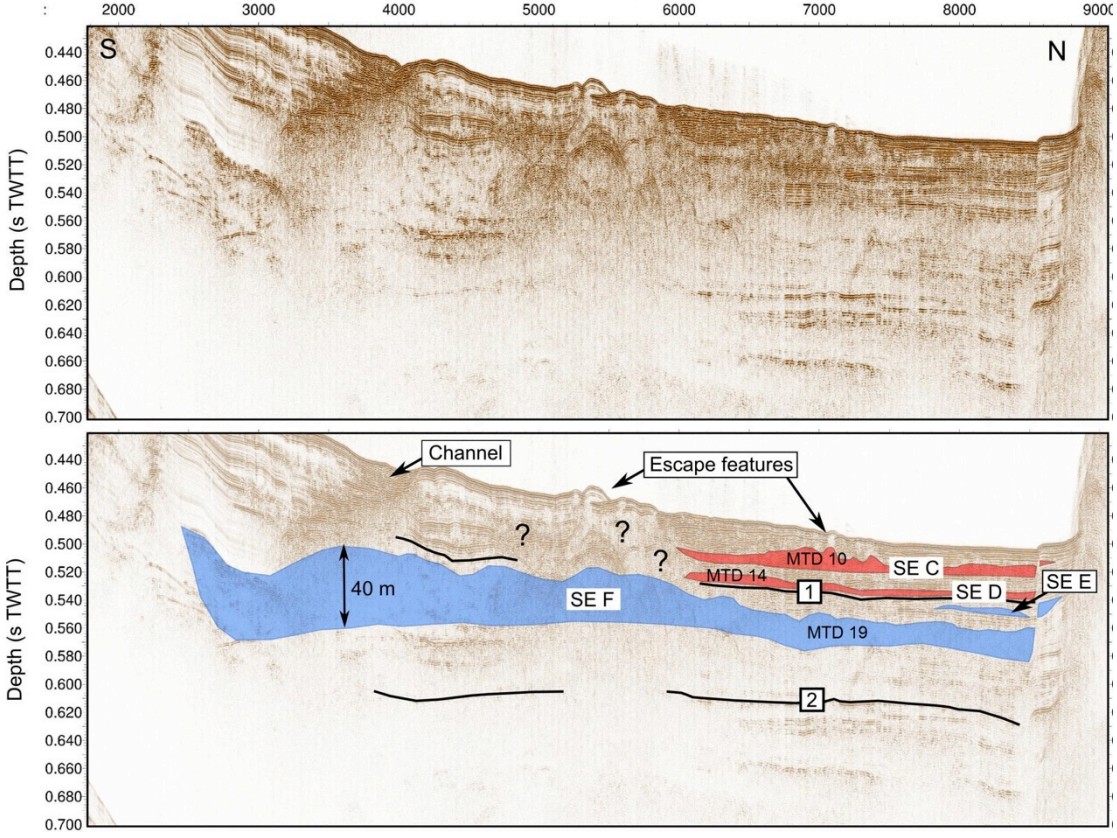


**Figure 5.** Examples of mass transport deposits in the Canyon basin. See the location of the Sparker seismic profile in Fig. 2

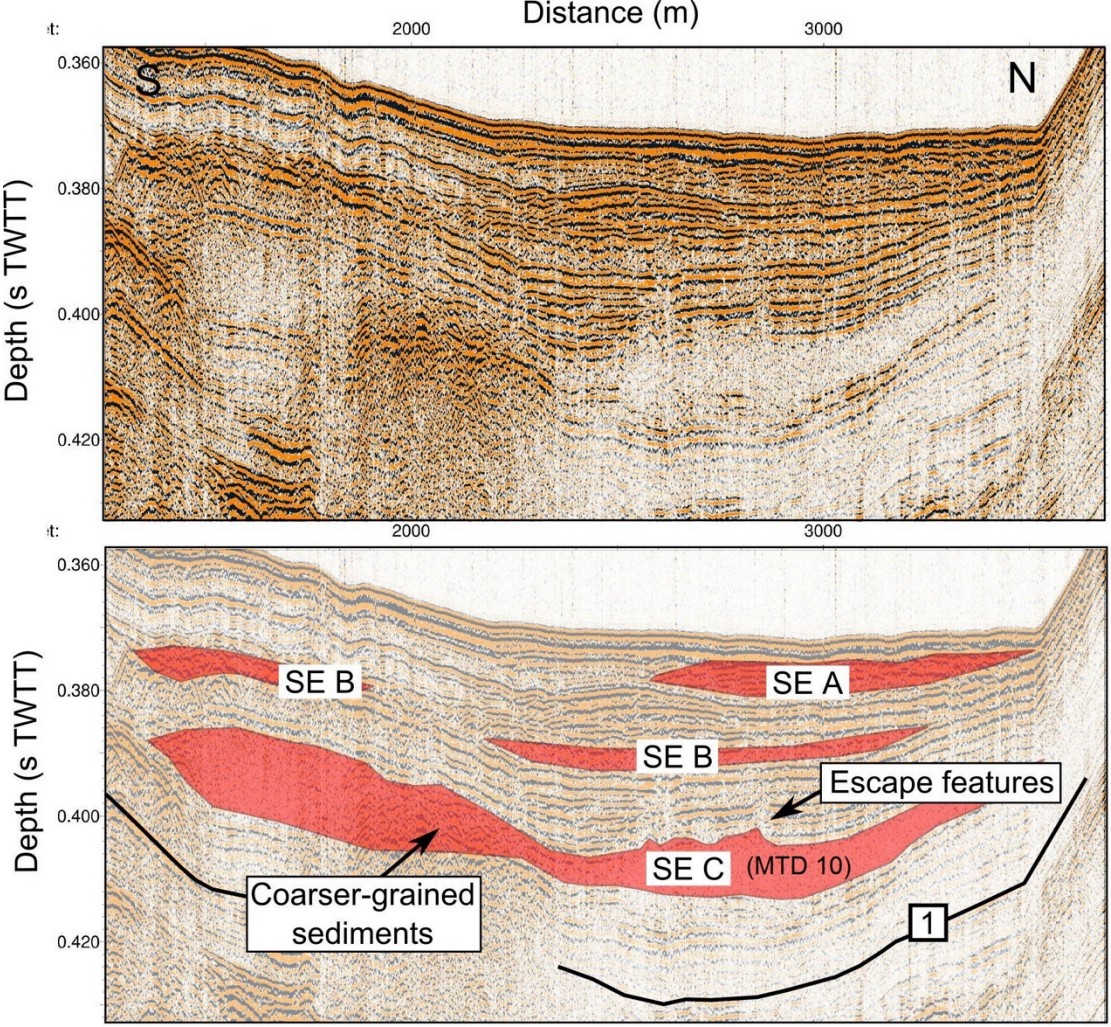

**Figure 6**. Volume distribution of the 22 MTDs studied in the Canyon and the Delphic Plateau basins. The names given to the three largest MTDs correspond to the notation in Fig. 2.

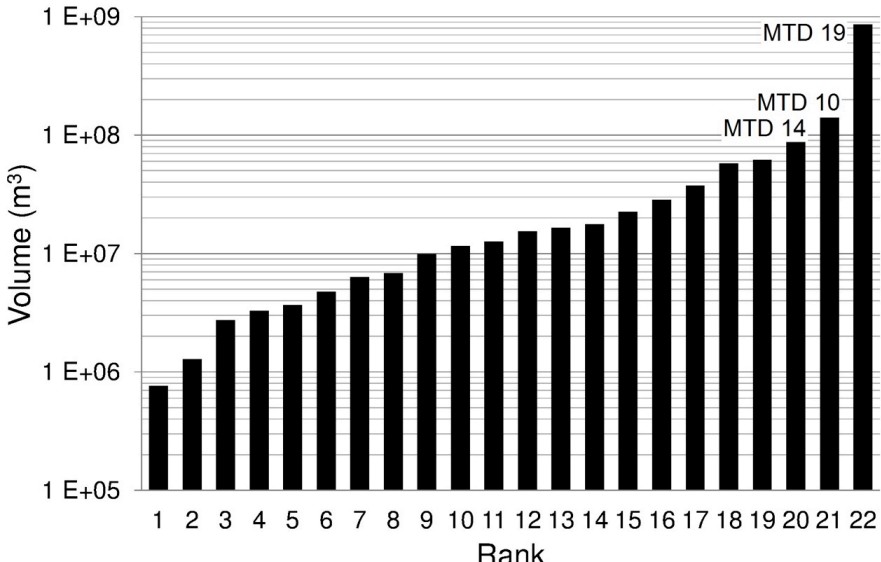

The stratigraphic position of MTDs in the Canyon and in the Delphic Plateau basins is not random. Most
of them are clustered and are defining multi-MTDs temporal "events", based on common un-deformed
underlying or overlying reflections that can be followed across the basin. Such correlations suggest that
six events of large clustered submarine mass wasting occurred over the last 130 ka. Two sliding events
(SE) are represented by clustered MTDs located between reflectors 2 and 1 (SE E and F). The four
others occurred during the Holocene: SE D comprises MTDs deposited just on top of the reflector 1, SE
C is located in the middle of the Holocene sequence, SE B somewhat higher, and finally SE A includes
MTDs at or near the sea floor responsible for its present-day hummocky topography. The spatial
distribution and the total volume of the MTDs associated to each of these events are represented in Fig.
238 2.
In some zones (Fig. 2), the existence or the geometry of MTDs is difficult to evaluate because of seismic
blanking and strong chaotic reflections affecting some stratigraphic intervals. Above reflector 1, the
stratigraphy is clear except regarding the southern extension of MTD 14 in SE D. The low amplitude,
almost transparent reflections characterizing the MTD deposit extends until a more chaotic and thicker
deposit associated with surface mounds  (Fig. 4). We could not decipher if the chaotic reflections that
disturb the seismic stratigraphy was associated with MTD 14 in SE D or in relation with sediment
remobilization from the underlying sliding event F (Figs. 4 and 5). So the mapped extension of MTD 14
in Fig. 2E is conservative and considered as a minimum. Below reflector 1, the amplitude of the
reflectivity sharply decreases, which is a characteristic of lowstand deposits in the Gulf (Bell et al.,
2008), and blanking occurs in two areas .In the Mornos Canyon area, a wide blanking area exists at a
depth of about 50 to 70 m below the sea floor, a few meters below reflector 1, in direct continuity with
the delta of the Mornos River. Blanking is thus a low-stand related feature and might correspond to
coarse grained, organic rich sediments of the Mornos River. Consequently, the stratigraphy of MTDs
between reflectors 2 and 1 is well established only below the Delphic Plateau. The other area associating
with blanking and strongly disturbed sediments forming mounds occurs at the junction between the
Mornos Canyon and the Delphic plateau at the foot of the Erineos foreset beds, at a depth similar to SE
F. Its origin is unknown, but it might be related to an MTD deposit in relation with MTD 19.
The definition of sliding events reflects a clustering of submarine landslides in a relatively short period
of time. It does not necessarily imply a synchronous occurrence of all submarine landslides included in
one event. Indeed, the accuracy of the correlation between separated MTDs that are interpreted to
belong to the same sliding event is in the order of one or two reflections in the seismic data. Deciphering
the exact MTD chronology within a sliding event was not possible because of the discontinuous
character of many reflections and the relatively large distance that separates some MTDs (up to 8.5 km).
This "stratigraphical" uncertainty corresponds to ~1-2 meters of sediment or, based on sedimentation
rate estimates, sliding events represent a set of MTDs that occurs over a period of 300 to 1000 years
(Lykousis et al., 2007).
Individual sliding events are characterized as follows (Fig. 2B to G):
*Sliding event A:*  Eight MTDs at or near the sea floor have been identified. Their spatial distribution
indicates that three of them result from slope failures in the Mornos delta and five from failures at
different locations along the southern margin (Fig. 2). The volumes of these MTDs range between ~4.7
$10^6$ m$^3$ and ~6.2 $10^7$ m$^3$, and the total volume of the eight MTDs is about ~2.2 $10^8$ m$^3$.
Some of these MTDs correspond to submarine landslides described in the literature (Galanopoulos
1964; Papatheodorou and Ferentinos 1997; Tinti et al., 2007). The MTD located north-east of the
Erineos delta results from a coastal landslide on this fan-delta in 1963, which triggered a large tsunami
on both sides of the Gulf (Galanopoulos et al., 1964; Stefatos et al., 2006). The MTD located at the foot
of the Meganitis fan-delta likely corresponds to a coastal landslide triggered by the 1995 Aigion
earthquake on this delta (Papatheodorou and Ferentinos 1997; Tinti et al., 2007). The volumes of
sediments involved in these two landslides have been estimated at ~4.6 $10^7$ m$^3$ from the data presented
by Stefatos et al. (2006), and about ~2.8 $10^7$ m$^3$ by Tinti et al. (2007), respectively. The corresponding
volumes estimated from the present study are ~6.1 $10^7$ m$^3$ and ~2.2 $10^7$ m$^3$, which are in the same order

of magnitude. Another well preserved but undated landslide headscarp has been identified by Tinti et al. (2007) on the eastern side of the Mornos fan-delta (Fig. 2). These authors estimated the volume of the sliding mass at ~9 $10^6$ $m^3$. Our data show a MTD located about 1 km downslope of the scarp, with an estimated volume of ~9.9 $10^6$ $m^3$ that fits remarkably well with the volume derived from the geometry of the scarp.

*Sliding event B:* The sliding event B comprises three MTDs located at the western tip of the canyon. They are located between 12 and 16 m below the sea floor and are relatively thin (~2 to 5 m thick) (Fig. 5). Location and geometry of the MTDs indicate that they result from slope failures in the Mornos fan-delta and in the Psathopyrgos scarp. The total volume of these MTDs is about ~1.7 $10^7$ $m^3$.

*Sliding event C:* The sliding event C includes one large MTD extending over a wide area below the Mornos Canyon and a part of the Delphic Plateau (MTD 10), and smaller deposits located at the foot of the southern slopes, in the Psathopyrgos scarp and Erineos fan-delta areas. The thickness of MTD 10 is shown in Fig. 7. Five local maxima are connected by a 2-5 m thick sheet of low-amplitude incoherent reflections. The thickest sediment accumulation (30 m) is located at the foot of the Erineos fan-delta. The other maxima are 5 to 10 m thick. Two are located at the western tip of the MTD and suggest sediment inputs from the Mornos fan-delta area and from the Psathopyrgos scarp (Fig. 5). The last two maxima are located in the south-eastern part of the deposit, with a possible source in the Erineos fan-delta (Fig. 7). The total volume that failed during sliding event C is about ~2.0 $10^8$ $m^3$, including ~1.4 $10^8$ $m^3$ for MTD 10.

**Figure 7.** Thickness of the largest MTDs deduced from the interpretation of Sparker seismic profiles with probable sediment paths indicated by red arrows (bold arrow: main sources). Contours represent the sea floor bathymetry interpolated from the Sparker data (one line every 20 m). Left: MDT 10 in sliding event C, the largest MDT from the sliding event C. Center: Thickness of MDT 14, the largest of the two MTDs that define the sliding event D. Bottom: The largest MTD from the presented inventory (MTD 10, sliding event F). The black bold lines represent two landslide head scarps likely linked to the MTD. The dotted line shows the location of the seismic profile in Fig. 8

314

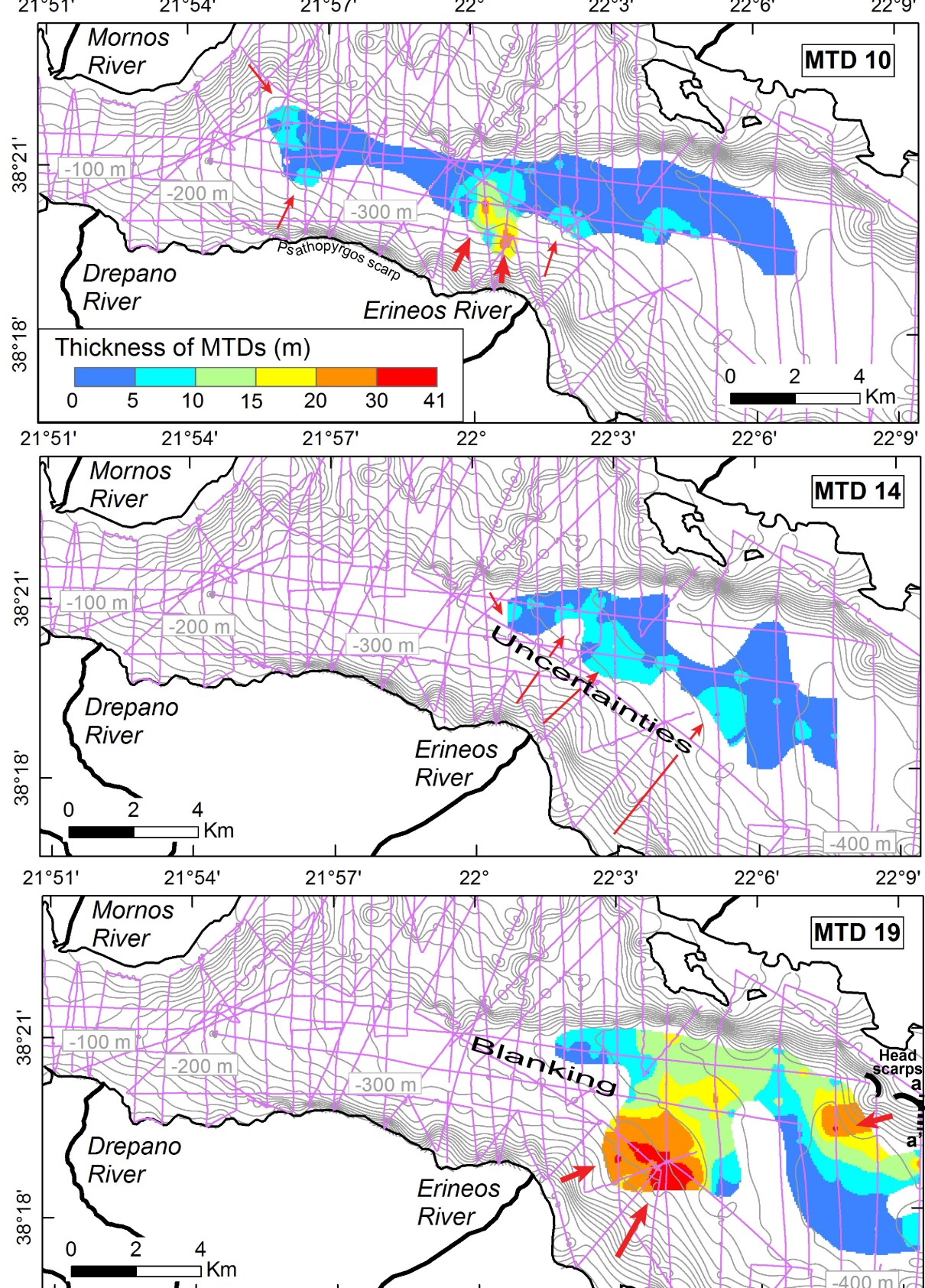

315
316

The geometry of MTD 10 suggests that slope failures occurred simultaneously in different parts of the westernmost gulf during sliding event C. The main source of sediment was the Erineos fan-delta, as attested by the location of the thickest sediment accumulation in the MTD 10, and by the presence of other MTDs at the same stratigraphic level between MTD 10 and the Erineos fan-delta (Fig. 2D).

*Sliding event D:* Two MTDs are located just on top of reflector 1 and define the sliding event D. Both are between ~2 and 10 m thick and spread over several square kilometres in front of the Erineos and Meganitis fan-deltas. The southern limit of the deposits is unclear, because the stratigraphy in the area between the two MTDs and the Erineos pro-delta is poorly constrained (hatching on Fig. 2E and question marks in Fig. 4). In this area, it is not sure whether the incoherent reflections located south of the SE D MTD at a similar depth represent the same MTD or the underlying, older (SE F), MTD or escape features from the latter, as suggested by the escape features observed at the sea floor (Fig. 4).

The isopach map of the largest deposit (MTD 14) is shown in Fig. 7 and suggests that it was fed by slope failure(s) mostly south of the Delphic Plateau probably from the Erineos Delta Fan. The volume of MTD 14 is estimated at ~$8.7 \ 10^7 \ m^3$, and the total volume of SE D MTDs is about ~$1.0 \ 10^8 \ m^3$. Considering uncertainties on the geometry of these MTDs' southern edges, these values are minimum estimates.

*Sliding event E:* Two MTDs define this sliding event. The largest one is located in the Delphic Plateau basin, just south of the Trizonia Island and has a volume of ~$6.6 \ 10^6 \ m^3$. The second is much smaller (~$1.3 \ 10^6 \ m^3$) and is located in the Canyon basin. Stratigraphically, both are located a few meters below reflector 1. However, they are horizontally 8.5 km apart, making the correlation uncertain. The total volume of the two MTDs in sliding event E is ~$7.9 \ 10^6 \ m^3$.

*Sliding event F:* The sliding event F is defined by one single large complex MTD (MTD19) (Fig. 2). This deposit is located in the Delphic Plateau basin. Stratigraphically, it belongs to the upper part of the unit between reflectors 2 and 1, suggesting that this event occurred during the last glacial period. With a volume of ~$8.6 \ 10^8 \ m^3$, this deposit is the largest MTD of the present inventory. It covers an area of 41 km$^2$, i.e., almost the whole Delphic Plateau. The isopach map reveals a main up to 50 m-thick sediment accumulation in the south-western part of the deposit (Fig. 4) and another ~30 m-thick depocenter in the north-eastern part (Fig. 7). The MTD is imaged as low amplitude, almost transparent chaotic reflections except in the thickest part where high-amplitude reflections could indicate coarser-grained sediments and locally preserved layering (Fig. 4). No sedimentological structure has been observed in the seismic profiles between the two maxima in thickness.

The geometry of the deposit and the absence of clear structure between the two depocenters support the idea of at least two simultaneous slope failures having generated this large MTD. The largest failure occurred south of the MTD, on the Meganitis or the Erineos fan-delta slopes. Considering the large volume of sediments in the south-western part of the MTD, we expected a major scar across the southern slopes, which we could not retrieve however neither from the seismic data, nor from published bathymetries (Lykousis et al., 2009; Nomikou et al., 2011, see our Fig. 1). Indeed, dozens of small head scarps and gullies dissect the slopes of the offshore Erineos and Meganitis deltas, making difficult the identification of large features. The second depocenter occurs near the north-eastern edge of the Delphic Plateau Basin, and upslope two submarine landslide headscarps located 2 km from each other were evidenced in seismic profiles (bold lines in Fig. 7). Cut through stratified hemipelagites, they are 11 and 15 m-high and are located at 300 and 195 m below the sea level, respectively (Fig. 8). Although it is not possible to reconstruct the 3D geometry of a single large headscarp from the seismic data, this would be a good candidate source of the thick sediment accumulation in the north-eastern part of MTD 19.

**Figure 8.** Sparker seismic profile illustrating a submarine landslide head scarp that is probably linked to the MTD 19. See the location of the profile in Fig. 7.

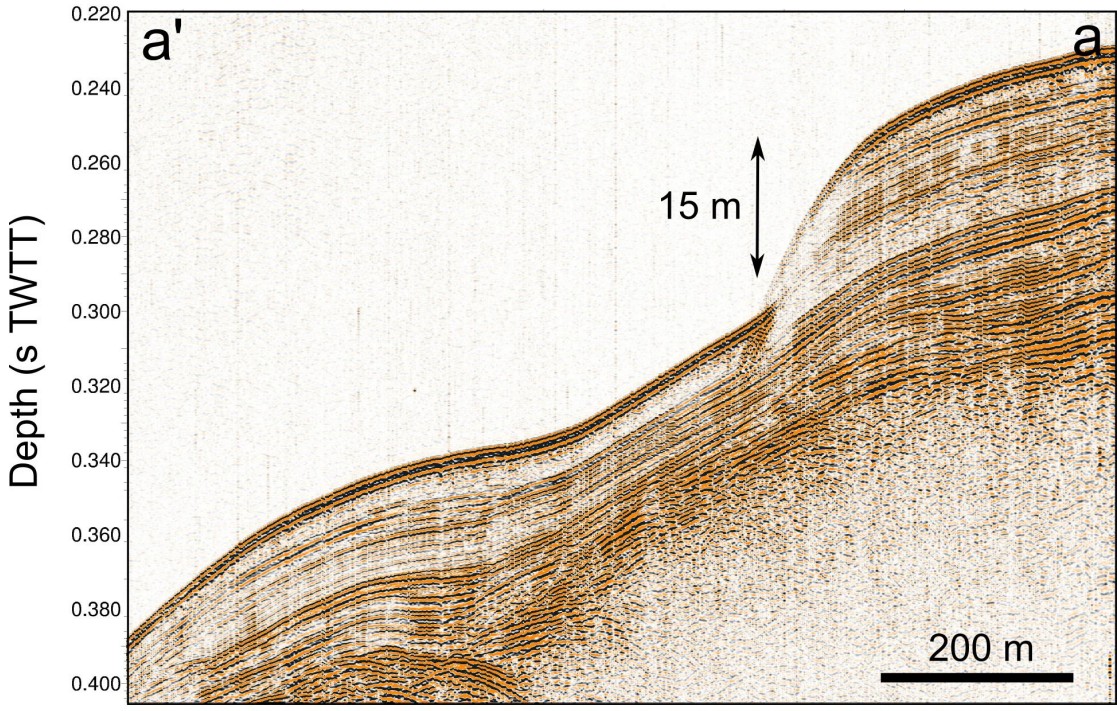

**5 Discussion**

*5.1 Limitations of the analysis*

Before discussing the implications of the presented MTD inventory in the deep flat basin in terms of sediment sources and triggering mechanisms, it is necessary to point out that only submarine landslides that have remobilized a sufficient quantity of sediments down to the basin floor are considered here. Moreover, the high-resolution seismic profiling system used does not permit identifying MTDs thinner than ~1 m. Consequently, our inventory is incomplete and could be refined by the use of very-high resolution seismic profiling systems and long cores.

*5.2 Sediment sources*

According to the mapping of the thickness of the deposits, large sliding events in the westernmost Gulf of Corinth mainly result from slope failures in, or close to, the Gilbert-type fan-deltas. Large sediment volumes were trapped in these deltas during the Holocene. As shown in Figure 1, Holocene foreset beds reach 40 to 60 m in thickness on average in the Eroneos and Meganitis fan-deltas, and sediment accumulation during the Holocene exceeding 100 m have been observed locally in between. These are the sources of MTD 10 in sliding event C and MTD 14 in sliding event D. The remarkable amount of sediments delivered to the gulf of Corinth during the Holocene probably results from large volumes of sediments stored onland during the last glacial period that were mobilized from river floodplains and colluvial deposits to rivers deltas. Widespread soil erosion resulting from human deforestation and agriculture during the second half of the Holocene also contributed to increase sediment fluxes in this period. Similarly, the previous period considered here spanning ~130 ka to ~11 ka is also characterized by a large sediment accummulation with a pile of 60 to 120 m forming the delta fronts of the Erineos and Meganitis delta (Fig. 1). These sources are some of the main source of MTD 10 in sliding event F.

The seismic facies of most large MTDs also implies that they are likely composed mainly of fine-grained sediments, and seismic profiles across fan-delta area have shown that the pro-delta foresets are locally made of a thick accumulation of stratified fine-grained sediments. These fan-delta sediments are probably the main source of sediments for the largest MTDs (MTD 10, 14 and 19). However, some smaller MTDs seem to be made of coarser-grained sediments according to the seismic character (e.g., in

SEs A and B in the Mornos Canyon basin), suggesting failure also occurred in coarser-grained parts of the fan-deltas located at the junction between the topset and the foreset beds (e.g., the 1963 slide in the Erineos fan-delta).

*5.3 Significance of the sliding events*

The data suggest that large submarine landslides have been triggered during six short periods of time over the last 130 ka. These sliding events include variable numbers of clustered MTDs, from one (SE F) to 8 (SE A). During three sliding events (C, D, F), a particularly large MTD accumulated at the basin floor, and it has been shown that these large MTDs resulted from several possibly synchronous slope failures. Similar MTD distributions have been observed in lakes in the Alps and in the Chilean Andes (Strasser et al., 2013; Moernaut et al., 2007). In these studies, the correlation of MTDs into a same "sliding event" was supported by radiocarbon dating and a simultaneous triggering has been proposed. Correlations between the mass wasting records of neighbour lakes and the historical seismicity revealed that most of these "sliding events" had been triggered by large earthquakes (Strasser et al., 2006; Moernaut et al., 2007). In the westernmost Gulf of Corinth, neither coring, nor dating is available to confirm our correlations between MTDs. Moreover, the occurrence of frequent turbidity currents (Heezen et al., 1966; Lykousis et al., 2007a) and small-scale submarine landslides perturbs the sediment layering and induces discontinuities in the seismic reflections, which makes MTD correlations based on the seismic stratigraphy less accurate there than in many lakes.

The case of sliding event A demonstrates that MTDs grouped within the same event did not necessarily occur at the same moment. Indeed, direct observation has shown that one MTD of this event occurred in 1963 AD and another in 1995 AD. By contrast, the synchronicity of different submarine landslides has been suggested for SE C, D and F from the complex shape of the large MTDs they include. Though not a proof, this lends support to the hypothesis of a seismic trigger of these three sliding events.

Consequently, the sliding events defined in this study may represent two different situations. In a first case, they correspond to a period of time of 0.3 to 1 ka during which several submarine landslides of various origins occurred. The sliding event A is such a case, with the coastal landslide caused in the Meganitis delta area by the 1995 Aigion earthquake and an aseismic coastal landslide in the Erineos delta area in 1963. The second case refers to likely simultaneous submarine landslides originating from different slopes and forming a wide MTD of complex shape in the basin floor. An example of this case, which is proposed to be earthquake-triggered, is the sliding event F, with a single MTD of complex shape. Sliding events C and D possibly belong to this category as well. There is insufficient data to allow for the determination of the nature of the minor events B and E.

Two main questions arise from these observations.
- Is seismicity the only forcing of SEs C, D and F or could other triggers or pre-conditioning factors such as sediment supply and sea level change have influenced the system?
- What are possible trigger mechanisms and/or pre-conditioning factors responsible for a cluster of slope failures such as SE A?

Urlaub et al. (2013) make inferences about controls on triggers of submarine landsliding from the statistical analysis of the ages of 68 very large slides (> 1 km[3]) around the world. From a subset of 41 slides that occurred during the best documented last 30 ky, they show that the distribution of number of events per ky resembles a Poisson distribution, suggesting that large submarine mass wasting might be essentially random or, at best, that the global-scale signal for a climatic control, through either sea level or sedimentation rate changes, is incoherent (non-uniform response of continental slopes worldwide) or too weak to be expressed clearly with such a small sample size. They also note that, though strong earthquakes might represent a temporally random trigger at the global scale, most of the slides in their data set are located in low-seismicity passive continental margins (Urlaub et al., 2013). Here, we first investigate the possible role of earthquakes through a comparative analysis of the frequency of sliding events and earthquakes in the Gulf of Corinth area. Then, other potential controls will be discussed by comparing the age distribution of the largest sliding events with published data about changes in

sediment dynamics and marine conditions in the Corinth Rift area. Owing to the small number of events and high age uncertainties, which rule out statistical considerations, we provide only a qualitative analysis.

*5.4. The possible role of large earthquakes*

The last four sliding events occurred during the last 10-12 ka, at an average rate of one event every 2.5-3 ka. Only two sliding events have been detected between ca. 130 ka and 10-12 ka. This high Holocene frequency compared with the ~120 kyrs anterior period may be attributed to two factors. First it might be a bias because the seismic reflections corresponding to the last glacial period (110-12ka) are less clear (lower amplitude and lower continuity) than the reflections from the Holocene interval. Consequently, medium-sized landslides such as those detected in SEs A and B might have been missed in the seismic unit between reflectors 2 and 1. Second, it could be attributed to a change in earthquake frequency due to a Holocene acceleration of the strain rates that was evidenced by fluvial morphometry (Demoulin et al., 2015) and subsidence markers (Beckers, 2015).

The average recurrence interval for large earthquakes (Mw 6-7) has been estimated in the central part of the Gulf of Corinth at ~500 yr during the Holocene, and ~400 yr for the period 12-17 ka, based on the record of "homogenites" in the deepest part of the Gulf (Campos et al., 2013). In the western Gulf of Corinth, estimates from palaeoseismological trenches on individual faults suggest an average recurrence interval ≤ 360 yr on the Aigion fault (Pantosti et al., 2004), and of 200-600 yr on the East Helike fault (McNeill et al., 2005) for the past 0.5-1 ka. It is clear, therefore, that large sliding events in the westernmost Gulf of Corinth were less frequent than Mw 6-7 earthquakes, during both the Holocene and the last glacial period. Consequently, while (anomalously?) large earthquakes could have triggered SEs C, D and F, as suggested above from the geometry of MTDs 10, 14 and 19, it is likely that other factors contributed to the occurrence of such large sliding events. These factors are explored in the next section.

*5.5 Other potential triggers and pre-conditioning factors*

Other possible processes that might have "pre-conditioned" or triggered sliding events in the Gulf of Corinth need to show a return period of at least 2.5 ka over the last 12 ka in order to fit the SE frequency. The following processes are proposed:
1. Sediment loading on top of a weak layer (e.g., gas-filled muddy sediments, as suggested for the area by Lykousis et al. (2009)) (pre-conditioning factor);
2. Pulses of increased onshore erosion inducing temporary increase of sedimentation offshore, in turn leading to slope overloading (pre-conditioning factor);
3. Sea level changes, which would have favoured slope failures during either lowstand conditions (Perissoratis et al., 2000) or sea level rises (Zitter et al., 2012) (pre-conditionning factor);
4. Changes in the circulation and/or intensity of bottom-currents progressively destabilizing submarine slopes through an increase in sedimentation or erosion rate (pre-conditioning factor);
5. Middle-term tectonic pulses, which would have temporarily increased the level of regional seismicity (Koukouvelas et al., 2005; Demoulin et al., 2015) (trigger);
6. Loading by exceptional storm waves (trigger);
7. Large supply of coarse-grained sediments at a river mouth during exceptional flooding events inducing slope failures by sediment overloading, as attested for the 1963 coastal landslide on the Erineos fan-delta by Galanopoulos et al. (1964) (trigger).

All these hypotheses are not directly testable. Moreover, it is likely that different pre-conditioning factors and triggers have interacted in various ways over the last 130 ka. Nevertheless, the four proposed pre-conditioning factors can be discussed by comparing the SE age distribution with independent data available for the region. We focus on the four events that mobilized a large volume of sediment ($\geq 10^8$ $m^3$, SEs A, C, D, and F) because they probably indicate slope failures in different parts of the westernmost Gulf, thus pointing to a regional signal. Even though these events have not been directly dated by coring, ages can be reasonably inferred from the seismic stratigraphy. The most recent sliding event (SE A) comprises MTDs at or near the sea floor and consequently occurred in the last 0.3-1 ka (a

range accounting for the thin layer of hemipelagites possibly covering some MTDs). Sliding event C likely dates from the Mid-Holocene (~6-7 ka) according to the Holocene age-depth curve in the central part of the Gulf of Corinth (Campos et al., 2013). The two MTDs defining SE D occurred just after the lacustrine to marine transition at the end of the Last Glacial, around 10-12 ka. Finally, the sliding event F dates from sometime in the last glacial period.

Among the listed pre-conditioning factors, onshore erosion dynamics in the Corinth Rift area is the best temporally documented. Fuchs (2007) presents the evolution of sedimentation rates in colluvial deposits on the southern shoulder of the Corinth Rift, in the Phlious Basin, 25 km south of Xylocastro, for the last 10 ka (Fig. 9). He identifies two main phases of land degradation between 6.5 and 8.5 ka, and from ~4 ka onwards. While the age of SE A corresponds to the end of the most recent period of land degradation, the much more uncertain age of SE C could correspond to the end of the land degradation phase at 6.5-8.5 ka (Fig. 9). The sliding event D is too old to be compared with the results of Fuchs (2007). In brief, a relation might exist between periods of high sediment supply from the watersheds and the occurrence of sliding events during the last 10 ky (hypotheses 1 and 2).

Less information is available about Late Pleistocene sediment dynamics in the area. Collier et al. (2000) suggest that the denudation rate at the eastern end of the Gulf in the Alkyonides Basin during the last glacial period (12-70 ka) was almost twice those of the Holocene and MIS 5 interglacials. Instead, six radiocarbon dates on long cores in the center of the Gulf of Corinth show a moderate increase in sedimentation rate between the end of the last glacial period (17- 12 ka) and the Holocene (Campos et al., 2013). Overall, these data suggest that the Last Glacial probably experienced the largest sedimentation rates over the last 130 ka in most of the Gulf of Corinth. This inference is however not valid at the western tip of the Gulf.  The comparison between isopach maps of the Holocene and the anterior 130-12 kyrs period evidences a large Holocene increase in sedimentation accumulation rate (Fig. 1). In the Delphic plateau basin, average sedimentation rate (excluding the thickness of MTDs) reaches ~2.4 mm/yr for the Holocene and ~ 0.4 mm/yr for the previous 120 kyrs. This is in line with the fact that only one large sliding event F was recorded during the ~60 ky-long Last Glacial. Increased sedimentation is thus a pre-conditioning factor of landsliding in the western Gulf..

Beside changes in erosion rates in the watersheds, the offshore realm underwent large changes between the last glacial period and today. From 70 to 12 ka, the Gulf of Corinth was a lake and the water level was around -60 m, assuming a constant depth of the Rion Sill over this period (Perissoratis et al., 2000). During this lowstand period, the extent of submarine slopes where submarine landslides can initiate were not significantly reduced, because the foreset beds of the Erineos and Meganitis that are the largest source of mass wasting sediments for the Delphic plateau extend down to the ~300 m isobaths.  The steepest slopes of these two prodeltas are located above isobaths -100m and between isobaths -150m and -200m according to the slope map of Nomikou et al. (2011), so unstable slopes above -60m that were submerged only in the postglacial period  cover a restricted area. At 10-12 ka, the rising waters in the Ionian Sea flooded the "Lake Corinth" through the Rion Sill (Moretti et al., 2003; VanWelden 2007). The sea level continued to increase from ca. -60 m to its present elevation until 5.5-6 ka, and bottom currents appeared in the study area (Beckers et al., 2016). The deposition of SE D occurred at 10-12 ka, when the water level started to increase in the Corinth Gulf. Water level change might change the stress field and pore pressure potentially affecting the earthquake cycle. Water level increase and bottom current initiation would also have favoured the destabilization of sediments deposited during the preceding glacial period. In the Sea of Marmara, observations by Zitter et al. (2012) and Beck et al. (2007) show an increase in large mass wasting events at the end of the last lacustrine period and at the beginning of the marine period that likewise can be explained by a change in oceanographic conditions, confirming the possible control of these pre-conditioning factors on SE D.

**Figure 9.** Comparison between the erosion dynamics over the last 10 ka from colluvial and alluvial archives in the Peloponnese (Fuchs, 2007), the rate of local water level changes, and the occurrence of large sliding

events in the westernmost Corinth Rift during the Holocene. Bars without error bars in the second panel
indicate minimum sedimentation rates.

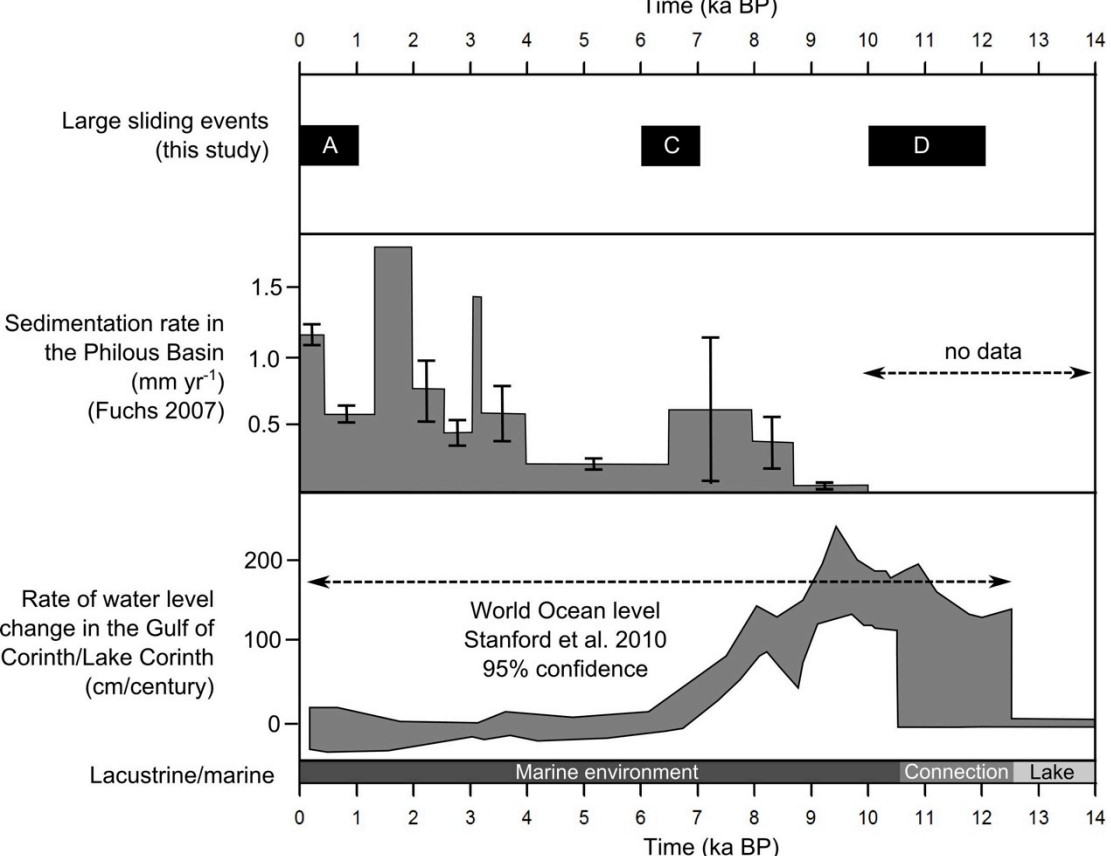

*5.6 Conceptual model for the sliding events*
Large sliding events (total volume $\geq 10^8$ m$^3$) occurred in the westernmost Gulf of Corinth with fairly
long recurrence intervals, $\geq$ 2.5 ka. We suggest that their temporal distribution is primarily controlled by
changes in pre-conditioning factors, which were a prerequisite for any landslide trigger to be effective.
In other words, the clustering of slope failures during distinct sliding events would depend on the
appropriate state of pre-conditioning factors, which occur only during limited periods of time. Two
types of pre-conditioning factors would have played a significant role, on one hand increased denudation
rates, identified at 17-70 ka, 6.5-8.5 ka and 0-4 ka and, on the other hand, dramatic changes in
oceanographic conditions that occurred at 10-12 ka. More generally, the SE frequency would reflect the
time needed to reload submarine slopes beyond their stability threshold after each event. Once the pre-
conditioning factor evolution has made the slopes prone to sliding, each individual sliding event is
characterized by either simultaneous submarine landslides producing large coalesced MTDs and
pointing to a likely seismic trigger (SEs C, D and F) or separate smaller slides caused by various lower-
intensity triggers (earthquakes, exceptional onshore flood events, as exemplified by the 1995 and 1963
coastal landslides, respectively) over a few centuries (SE A).
Finally, we underline that the sliding processes have not been clearly identified in this study. Lykousis et
al. (2009) mention debris flows and avalanches for slope failures on steep fan-delta slopes (2-6°) in the
western Gulf of Corinth, and rotational slumps on low angle (0.5-2°) prodelta slopes. One sharp head
scarp identified in this study also shows that at least one translational slide happened in hemipelagites
accumulated far from the main river outlets.
*5.7 Implications for tsunami hazard in the Gulf of Corinth*
Among the 32 MTDs identified in this study, MTD 19 stands out as a particularly large feature (a little
less than 1 km$^3$ in volume). This is 6 times the volume of the second largest MDT identified in this
study, and about two orders of magnitude larger than the range previously proposed for the size of
submarine landslides in the westernmost Gulf of Corinth (Lykousis et al., 2007). It is also 6 times larger
than the largest MTD reported in the rest of the Gulf of Corinth, which occurred in the area of the
Perachora Peninsula (Papatheodorou et al.,1993; Stefatos et al., 2006). MTD 19 likely resulted from the
coalescence of at least two probably synchronous major slides. If correct, these slides should have
triggered very large tsunamis waves, probably larger than those reported by historical sources in the
westernmost Gulf of Corinth, which were triggered by small to medium-sized slope failures
(Papadopoulos 2003; Stefatos et al., 2006; Tinti et al., 2007).
**6 Conclusion**
We documented the existence of large mass wasting events during the Holocene and the Late
Pleistocene in the westernmost Gulf of Corinth. Mass wasting events consist in submarine or coastal
landslides that occurred during short periods of time. Six large mass wasting events are listed, their
associated deposits locally representing 30% of the sedimentation since 130 ka in the Delphic Plateau
Basin. In the case of large MTDs (up to almost 1 km$^3$ for the largest), a simultaneous triggering of
separate slope failures is proposed, suggesting a seismic origin. However, it is suggested that the
temporal distribution of sliding events is primarily controlled by the evolution of pre-conditioning
factors. Two main pre-conditioning factors are identified, namely (1) the time needed to slope reloading
after an event, which varied in relation with temporally varying sedimentation rates, and (2) dramatic
changes in water depth and water circulation that occurred 10-12 ka ago during the last post-glacial
transgression. Finally, it is likely that these sliding events have triggered large tsunami waves in the
whole Gulf of Corinth, in some cases (much?) larger than those reported in historical sources.

Competing interests. The authors declare they have no conflict of interest.
Acknowledgement. This work has been funded within the ANR SISCOR project directed by Pascal
Bernard, at Institut de Physique du Globe (Paris) and by FNRS- Grant for Researchers (CC) ID
14633841. Arnaud Beckers's PhD grant was supported by the Belgian FRIA. Funding for Arnaud
Beckers' stays in the ISTerre Laboratory was provided by a grant from la Région Rhône-Alpes.  The
authors warmly acknowledge R/V ALKYON's crew, Koen De Rycker (RCMG), and Pascale Bascou
(ISTerre) for technical support, and the whole SISCOR scientific team for fruitful discussions. We
would like to thank the reviewers (David Tappin and anonymous) for their comments that improved the
paper.

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
