# Peer review of "CHARACTERISTICS AND FREQUENCY OF LARGE SUBMARINE LANDSLIDES AT THE WESTERN TIP OF THE GULF OF CORINTH"

_Natural Hazards and Earth System Sciences, 2017_

## Referee Comment (RC1) · Anonymous Referee #1 · 1 Jan 2018

The paper presents the interpretation of landslide deposits from different sets of single-channel seismic reflection profiles across the Gulf of Corinth. From a hazard perspective, evidence of mass transport complexes is important, particularly if these can be linked to the preconditioning and triggering factors. In this area, recurrence rate of landslides appears significant, and as landslides can generate destructive tsunamis, assessing the source areas, causes and consequences are important. This is a well-written paper, with a good data set and logical structure, even though the content is largely descriptive. There are nevertheless a few points that I am missing from the paper: Whereas identifying landslide deposits and obtaining the volumes involved are essential in a geohazard perspective, there is also a need to better define the land-

slide processes and consequences. - I am somewhat surprised to see that the source areas from the different landslide events remain very poorly constrained, despite the fact that some of the landslide deposits are quite large, and cover a significant part of the basin. - The preconditioning and triggering factors remain uncertain. I note that the point (abstract) of dramatic changes in water depth and water circulation at 10-12 ka is only applicable to a some of the cases. - Landslide dynamics and the tsunami potential are briefly mentioned but not really addressed. Such assessment would require modelling, but also information about the soil properties, the source areas, etc. Not all landslides will create tsunamis (see Løvholt et al., 2017). - The authors report landslide volumes, calculated from a (sparse) grid of seismic reflection profiles. The authors should mention the method used to obtain these values (e.g., gridding algorithm) as well as adding a statement about the uncertainty, particularly considering the line spacing of the seismic lines, and the lack of 3D seismic data. Can we be sure that the spatial extent mapped is a realistic impression of the failures or can they be over-estimated, due to the gridding and missing out areas where there are no deposits (but not evidenced because of the lack of data). This should be added as a key point under 5.1 Limitations of the analysis. - What is the onshore-offshore relationship of the landslides? - In the interpretation, the authors repeatedly refer to blanking but they do not really illustrate what is it and what the causes may be. - Likewise, the authors refer to coarser grained material in a deformed mass transport deposit, but there is no evidence for this. I doubt that one would be able to observe this from sparker data, as the masses are essentially deformed. Maybe speculation?

Smaller comments: - I would recommend making the seismic profiles with the same vertical exaggerations or same scales to facilitate comparison. Likewise, please add an indication on the figures where the seismic lines cross. - Terminology is in places confusing. I understand from this paper that landslide event actually refers to a certain interval in time (not specified) during which various landslides (with different source locations) may occur. Thus, different landslides compose a landslide event. - The map should contain all geographical references used in the text. This is currently not the

case. - On Figure 1, I would recommend adding a colour-coded (shaded relief or so) topographic/bathymetry map and slope map, as both are important to understand the processes. The maps should ideally cover the onshore and offshore part. Note that the "grey lines" referred to are not only the seismic grid but also bathymetric contour lines. Add the location of the Delphic Plateau, and the "Canyon". - There are a few typos in the text - Figure 2: explain the horizons [1] and [2] - The term "outcrop" suggests that something was eroded on top. This may not be the case for the youngest landslide deposits. Consider using exposed as the seafloor - Figure 6 is too small, and ideally, the maps should all use the same area, to facilitate comparison. This would be a good place to add the various source areas.

---

## Referee Comment (RC2) · D. Tappin (Referee) · 14 Feb 2018

Overview The paper describes the frequency and characteristics of small volume submarine landslides in the Gulf of Corinth over the past 130 ka. The landslides have the potential to generate hazardous tsunamis, with one historical event recorded. Potential landslide preconditioning and triggers are discussed. Six major landsliding events are recognised, of which three are relatively large volume. The age of the events are dated by their relationship to two regional seismic horizons interpreted as major flooding events and dated at 130 and 10-13ka. Most slide events (four) are identified as Holocene, with two others older than 10 to 12 ka. Although the volumes of most slides

are quite small (largest 1km3) one, in historical times (1995), generated a significant tsunami. The paper is a dense read, because of the number and complexity of the landslide failures and the relationships to triggering mechanisms and preconditioning. Comments The strengths of the paper are in the seismic data and its interpretation. The weakness is the lack of sample data to identify the sedimentology of the slides and their ages, which are based on the slides relationships to the two regional horizons. As with all submarine landslides, earthquakes are proposed as the most likely trigger. Earthquake records (from sediments) are confined to the past 17 thousand years, with frequencies of 400-500 years for the period 12-17ka in the central Gulf of Corinth and in the western Gulf (from palaeoseismology) 200 to 600 years. Preconditioning factors are identified from events in other regions outside the western Gulf. Focussing on the science, I am surprised that the dated regional horizons are not used more fully to understand sedimentation rates and the potential rates of sediment recharging in the western basin. These might better inform on the local differences between the glacial and post glacial environments that would influence slide failure. I am also surprised that there is not more consideration of the major difference between the glacial and Post glacial sea levels in the context of the slides and their headscarps. Consideration of Figure 1 suggests that lowering sealevel by 60 metres makes a major difference in some regions that may influence sedimentation and sliding. Whether the difference in sealevels is important or not, it would be informative to see the effects on a figure. It would also help the reader if some of locations of data which underpin the interpretations, which are outside the area were identified on a map. These include the Philious Basin (Page 13, lines 457-458) and the Alkyonides basin (Page 113, Line 467). Identifying the location of these would identify their relevance. The interpretation of the earthquake triggering of the landslides is undoubtedly reasonable, but the evidence is very sparse. It seems that only the 1995 earthquake triggered the MTD at the foot of the Meganitis fan, but this is hypothetical. What was the trigger of the 1963 landslide which caused a major tsunami, was it just sediment loading? The following discussion of the relationships between earthquake frequency and landsliding is also questionable, because it is assumed that the earthquake frequency for the glacial period is the same as for the Holocene (page 12, lines 416-418) which seems to me to be unlikely. This is quite a jump in the interpretation as it underpins much of the subsequent discussion on triggering and preconditioning – but that is always the problem with MTDs. I guess it doesn't invalidate the interpretations too much. Regarding my comment on the complexity of the paper, I make some suggestions. There are geographical names mentioned in the text which are not on the figures, e.g, Delphic Plateau, Canyon basin, possibly others. With regard to the organisation of the paper, I found it hard to understand the full setting of the GoC from the background sections because back ground material is distributed later in the paper. Other material which should be presented early on in the Background includes; the stratigraphic framework (Page 2 lines 90-95) and the palaeolake levels (Page 13 lines 477-485). Including these would provide a broader picture to background the environmental changes over the 130 ka time period. Conclusions Apart from my above comments, this is an interesting paper identifying the potential hazard from submarine landslides in an enclosed basinal area, where future events, if of sufficient volume would be a tsunami hazard. It is well organised and well written. The remote data set is good, the temporal controls on the events are weak, but the innovative approach, using the (sparse) data applicable to this, results in a plausible story which should be published with some modification.

David Tappin 14th February 2018

---

## Author Comment (AC1) · 5 Mar 2018

Comments of referee 2: Overview The paper describes the frequency and characteristics of small volume submarine landslides in the Gulf of Corinth over the past 130 ka. The landslides have the potential to generate hazardous tsunamis, with one historical event recorded. Potential landslide preconditioning and triggers are discussed. Six major landsliding events are recognised, of which three are relatively large volume. The age of the events are dated by their relationship to two regional seismic horizons interpreted as major flooding events and dated at 130 and 10-13ka. Most slide events (four) are identified as Holocene, with two others older than 10 to 12 ka.

[Figure]

Although the volumes of most slides are quite small (largest 1km3) one, in historical times (1995), generated a significant tsunami. Author's response: Note that there was another significant tsunami generated during historical times by one of the slide we have mapped: the 1963 tsunami was generated by a submarine landslide on the Erineos prodelta. Moreover, numerous other historical tsunamis have been reported in the western Gulf of Corinth: 1996, 1984, 1965, 1888, 1887, 1861, 1817, etc. (the full list is in Papadopoulos, 2003, Natural Hazard; summary in Beckers et al., 2017, Marine Geology)

Comments of referee 2: The paper is a dense read, because of the number and complexity of the landslide failures and the relationships to triggering mechanisms and preconditioning.

Comments of referee 2: The strengths of the paper are in the seismic data and its interpretation. The weakness is the lack of sample data to identify the sedimentology of the slides and their ages, which are based on the slides relationships to the two regional horizons. As with all submarine landslides, earthquakes are proposed as the most likely trigger. Earthquake records (from sediments) are confined to the past 17 thousand years, with frequencies of 400-500 years for the period 12-17ka in the central Gulf of Corinth and in the western Gulf (from palaeoseismology) 200 to 600 years. Preconditioning factors are identified from events in other regions outside the western Gulf.

Author's response: We fully agree with the referee about the paper weakness. There are no long coring across the western gulf of Corinth, to establish a relationship between MTDs, their sedimentological imprint and an earthquake catalogue.

Comments of referee 2: Focussing on the science, I am surprised that the dated regional horizons are not used more fully to understand sedimentation rates and the potential rates of sediment recharging in the western basin. These might better inform on the local differences between the glacial and post glacial environments that would influence slide failure.

Author's response: We have now added new data in figure 1 showing isopach 1 of the Holocene and of the previous glacial interglacial period in answer to referee 1, but we have not fully exploited the data to discuss the potential influence of sediment recharging in the western basin. In the discussion, we now stress that there is a large difference regarding the sediment recharging between the Holocene Period and the previous 130-12kyrs one. Considering the spot between the Erineos and the Meganitis delta-fans (See Fig. 1) where we could have a reliable record of sediment accumulation over the last 130 kyrs, up to $\sim$ 100 m were accumulated over the last 10.5 to 12.5 kyrs and up to 125 m were accumulated over the previous $\sim$120 kyrs period. So there is an order of magnitude difference in sediment recharging between the Holocene and the previous period. We are now discussing these facts in the text.

Author's changes in manuscript: Change in the Discussion section 5.5 Other potential triggers and pre-conditioning factors

...Overall, these data suggest that the Last Glacial probably experienced the largest sedimentation rates over the last 130 ka in most of the Gulf of Corinth. This inference is however not valid at the western tip of the Gulf. The comparison between isopach maps of the Holocene and the anterior 130-12 kyrs period evidences a large Holocene increase in sedimentation accumulation rate (Fig. 1). In the Delphic plateau basin, average sedimentation rate (excluding the thickness of MTDs) reaches $\sim$2.4 mm/yr for the Holocene and $\sim$ 0.4 mm/yr for the previous 120 kyrs. This is in line with the fact that only one large sliding event F was recorded during the $\sim$60 ky-long Last Glacial. Increased sedimentation is thus a pre-conditioning factor of landsliding in the western Gulf.

Comments of referee 2: I am also surprised that there is not more consideration of the major difference between the glacial and Post glacial sea levels in the context of the slides and their headscarps. Consideration of Figure 1 suggests that lowering sea level by 60 metres makes a major difference in some regions that may influence sedimentation and sliding. Whether the difference in sea levels is important or not, it would be informative to see the effects on a figure.

Author's response: We agree that the lowering sea level might make a difference in some part of the Gulf but not a major one regarding the delta-front bordering the southern edge of the western Corinth Gulf. The new figure 1 showing the high resolution bathymetry of Nomikou et al. (2011) clearly show that the isobaths -100 is located close to the shoreline all along the faulted southern coast west of Aigio, and that the foreset beds extend to isobaths -300m. So the submarine slopes where submarine landslides can initiate are not significantly different between the Last Glacial Period and the postglacial period, they might be a little more restricted during the Last Glacial Period.

Author's changes in manuscript: The following clarification was included in the Discussion section and the subsection 5.5 Other potential triggers and pre-conditioning factors . . . During this lowstand period, the extent of submarine slopes where submarine landslides can initiate were not significantly reduced, because the foreset beds of the Erineos and Meganitis that are the largest source of mass wasting sediments for the Delphic plateau extend down to the ∼300 m isobaths. The steepest slopes of these two prodeltas are located above isobaths -100m and between isobaths -150m and -200m according to the slope map of Nomikou et al. (2011), so unstable slopes above -60m that were submerged only in the postglacial period cover a restricted area. . . .

Comments of referee 2: It would also help the reader if some of locations of data which underpin the interpretations, which are outside the area were identified on a map. These include the Philious Basin (Page 13, lines 457-458) and the Alkyonides basin (Page 113, Line 467). Identifying the location of these would identify their relevance.

Author's response: We fully agree with the reviewer that the reader needs to know were

the locations of data, which underpin the interpretations need to be added. However the paper is already long with 9 figures and we did not want to include a new one. So we choose to indicate clearly in the text the relevant information regarding the location of the data with respect to the Gulf of Corinth.

Author's changes in manuscript: in the Discussion section and the subsection 5.5 Other potential triggers and pre-conditioning factors Regarding the location of the Alkyonides Basin, we add the clarification that it is located at the eastern tip of the Gulf in the following way: "Collier et al. (2000) suggest that the denudation rate at the eastern end of the Gulf in the Alkyonides Basin" Regarding the location of the Philious Basin we did the same: "Fuchs (2007) presents the evolution of sedimentation rates in colluvial deposits on the southern shoulder of the Corinth Rift, in the Philious Basin, . . ."

Comments of referee 2: The interpretation of the earthquake triggering of the landslides is undoubtedly reasonable, but the evidence is very sparse. It seems that only the 1995 earthquake triggered the MTD at the foot of the Meganitis fan, but this is hypothetical. What was the trigger of the 1963 landslide which caused a major tsunami, was it just sediment loading? The following discussion of the relationships between earthquake frequency and landsliding is also question able, because it is assumed that the earthquake frequency for the glacial period is the same as for the Holocene (page 12, lines 416-418) which seems to me to be unlikely. This is quite a jump in the interpretation as it underpins much of the subsequent discussion on triggering and preconditioning – but that is always the problem with MTDs. I guess it doesn't invalidate the interpretations too much.

Author's response: There are no relationship between the earthquake frequency and climatic changes. The earthquake cycle is linked to loading on faults due to geodynamic processes at depth independent of surface processes. So indeed we assumed that earthquake frequency is nearly constant. There are still some potential effects on the seismicity due to the rapid water level changes at the beginning of the Holocene, which would have change the stress field and the pore pressure. But nobody has modeled it, and its effects on the fault system in the Gulf of Corinth are unknown. Any lenghly discussion about it would be very speculative. We still include a short sentence to take into account the referee remark.

Author's changes in manuscript: in the Discussion section and the subsection 5.5 Other potential triggers and pre-conditioning factors

"The deposition of SE D occurred at 10-12 ka, when the water level started to increase in the Corinth Gulf. Water level change might change the stress field and pore pressure potentially affecting the earthquake cycle. Water level increase and bottom current initiation would also.."

Author's response: The earthquake frequency might also have changed because of change in the geodynamics of the rift over the 130 kyrs timescale. Previous studies (i.e. Ford, M., et al. "Rift migration and lateral propagation: evolution of normal faults and sediment-routing systems of the western Corinth rift (Greece)." Geological Society, London, Special Publications 439.1 (2017): 131-168) suggest a rapid evolution with respect to fault growth and linkage. Demoulin et al. (2015) also evidenced an Holocene acceleration of the strain rates using fluvial morphometry and we also found a large increase in Holocene subsidence rate compared to the previous period probably linked with an acceleration of the deformation (Beckers, A., 2015, Late quaternary sedimentation in the western gulf of Corinth: interplay between tectonic deformation, seismicity, and eustatic changes, PhD thesis, pp. 260), but these later finding will be independently published in a peer-review journal. Given the uncertainties about the earthquake frequency, we choose not to have an extended discussion about the topics, because it would be too speculative. We still have included the following changes in the text

Author's changes in manuscript : in the Discussion section and the subsection 5.4. The possible role of large earthquakes "This high Holocene frequency compared with the ∼120 kyrs anterior period may be attributed to two factors. First it might be a bias, because the seismic reflections corresponding to the last glacial period (110-12ka) are less clear (lower amplitude and lower continuity) than the reflections from the Holocene interval. Consequently, medium-sized landslides such as those detected in SEs A and B might have been missed in the seismic unit between reflectors 2 and 1. Second, it could be attributed to a change in earthquake frequency due to a Holocene acceleration of the strain rates that was evidenced by fluvial morphometry (Demoulin et al., 2015) and subsidence markers (Beckers, 2015)."

Comments of referee 2: Regarding my comment on the complexity of the paper, I make some suggestions. There are geographical names mentioned in the text, which are not on the figures, e.g, Delphic Plateau, Canyon basin, possibly others.

Author's response: We made modifications to mention the names in the figures already for referee 1.

Comments of referee 2: With regard to the organisation of the paper, I found it hard to understand the full setting of the GoC from the background sections because back ground material is distributed later in the paper. Other material which should be presented early on in the Background includes; the stratigraphic framework (Page 2 lines 90-95) and the palaeolake levels (Page 13 lines 477-485). Including these would provide a broader picture to background the environmental changes over the 130 ka time period.

Author's response : We already change the setting section to provide a clearer picture according to the remarks of referee 1, but we now include a new paragraph in the setting to provide more information also regarding the stratigraphic framework and the palaeolake levels.

Author's changes in manuscript : The following paragraph was added to the setting section: The shallow sedimentary infill of Gulf of Corinth infill consists of a distinct alternation between seismic-stratigraphic units with parallel, continuous high-amplitude reflections and units with parallel, continuous low amplitude reflections to acoustically transparent seismic facies (e.g. Bell et al., 2008; Taylor et al., 2011). Generally, the semi-transparent units are thicker than the highly reflective units (e.g. Taylor et al., 2011). These alternating seismic-stratigraphic units have been observed throughout the Gulf of Corinth and have been interpreted as depositional sequences linked to glacio-eustatic cycles (Bell et al., 2008; Taylor et al., 2011). Because of the presence of the 62 m deep Rion Sill at the entrance of the Gulf, the Gulf of Corinth was disconnected from the World Ocean during Quaternary lowstands and was thus a non-marine sedimentary environment. The marine and non-marine environments are associated with different climatic regimes (e.g. Leeder et al., 1998). During glacial stages, the sparse vegetation cover was more favourable to erosion than during interglacials, so high quantities of sediments were routed towards the Gulf (Collier et al., 2000). These lowstand deposits appear as thick, low-reflective units. The thin, high-reflective units are interpreted to represent the marine highstand deposits. The last lacustrine-marine transition has been sampled in different sedimentary cores (Collier et al., 2000; Moretti et al., 2004; Van Welden, 2007; Campos et al., 2013).

Conclusions Apart from my above comments, this is an interesting paper identifying the potential hazard from submarine landslides in an enclosed basinal area, where future events, if of sufficient volume would be a tsunami hazard. It is well organized and well written. The remote data set is good, the temporal controls on the events are weak, but the innovative approach, using the (sparse) data applicable to this, results in a plausible story which should be published with some modification.

Please also note the supplement to this comment:
https://www.nat-hazards-earth-syst-sci-discuss.net/nhess-2017-371/nhess-2017-371-AC1-supplement.pdf
* * *
[Figure]

**Fig. 1.**

[Figure]

**Fig. 2.**

**Supplement:**

[revised manuscript text omitted]

---

## Author Comment (AC2) · 5 Mar 2018

Comments from Referee 1: The paper presents the interpretation of landslide deposits from different sets of single- channel seismic reflection profiles across the Gulf of Corinth. From a hazard perspective, evidence of mass transport complexes is important, particularly if these can be linked to the preconditioning and triggering factors. In this area, recurrence rate of landslides appears significant, and as landslides can generate destructive tsunamis, assessing the source areas, causes and consequences are important. This is a well- written paper, with a good data set and logical structure, even though the content is largely descriptive. There are nevertheless a few points

that I am missing from the paper: Whereas identifying landslide deposits and obtaining the volumes involved are essential in a geohazard perspective, there is also a need to better define the land- slide processes and consequences. - I am somewhat surprised to see that the source areas from the different landslide events remain very poorly constrained, despite the fact that some of the landslide deposits are quite large, and cover a significant part of the basin.

Author's response: Considering this question of the reviewer and some of the following ones, we realized that part of the context regarding the geohazard landslide perspective is missing. First, we talk about earthquakes and landslides triggered by earthquakes, but did not provide a fault map. We presently add a new set of figures labeled Figure 1. At the top of Figure 1, we now display the active faults with the high resolution bathymetry obtained by Nomikou et al. (2011). Second, another element was also missing. Readers without previous knowledge of the submarine context of the Gulf of Corinth would not realize that very large amount of uncompacted sediments are available in steep submarine delta slopes, which is a preconditioning factor. In the setting (line 70-74) we mention that "the western gulf is bordered to the south by 400 m high Gilbert deltas built by the Erineos, Meganitis and Slinous river, and at its north-western end, by the fan delta of the Mornos River. . .. The delta fronts are highly unstable (..". But the comments of the reviewer show that a more precise context is necessary.

Author's changes in manuscript: We now show in Figure 1 in addition to the active faults and the submarine bathymetry, which evidences the steep and wide delta fronts, the morphosedimentary map of Holocene deposits and the isopach maps of the Holocene and the previous glacial-interglacial period.

Author's response: The isopach maps evidence the very large volume of sediments accumulated on steep unstable slopes that is available for mass transport. Most landslide deposits documented in paper have sources in these steep overloaded delta fans located along the southern coast and at the north-western end of the Gulf. So in fact the source areas are broadly very well defined. However given that there are high quality multi-beam data available but only a high-resolution raster map of the bathymetry by Nomikou et al. (2011) it is not possible to define in more details the source areas.

Author's changes in manuscript: We change the setting section in the following way to provide clearer indication about the morphological setting and the inferred source areas located in steep slopes surrounding the flat basin. " The western Gulf of Corinth is characterized by a relatively flat deep basin dipping gently to the east. Featuring a narrow canyon in the west, it widens in the east (Delphic Plateau, Fig. 1). It is bordered by steep slopes on all sides (Fig. 1) To the north, it is limited by the Trizonia scarp with slopes ranging from 25° to locally more than 35° and the associated Trizonia Fault (Nomikou et al., 2011); these slopes are mostly devoid of sediments which are trapped in the bay areas to the north (Fig. 1B). To the south, the western Gulf is bordered by 400m high Gilbert deltas built by the Erineos, Meganitis and Selinous rivers that lie in front of the active Psathopyrgos, Kamari and Aigion Faults running along or near the coastline. Delta fronts have 15° to 35° slopes incised by gullies (Lykousis et al., 2007; Nomikou et al, 2011) and consist of a thick pile of fine grained sediments. The delta-front sediments accumulated over the Holocene and the previous glacial-interglacial period have thicknesses, respectively, larger than 50m and 100 m (Fig. 1B and 1C; Beckers, 2015; Beckers et al, 2016). At the north-western end of the Gulf, lies the largest fan-delta of the Mornos River that drains 913 km2 and is by far the largest watershed among the rivers flowing toward the westernmost Gulf of Corinth (Fig. 1A). The delta fronts are highly unstable (Ferentinos et al, 1988; Lykousis et al., 2009), which favours frequent submarine landsliding (Stefatos et al., 2006; Tinti et al., 2007; Fig. 1B). During the last centuries, submarine landslides have been triggered by earthquakes and by sediment overloading on steep slopes (Galanopoulos et al., 1964; Heezen et al., 1966). Numerous debris-flow deposits and mass-transport deposits (MTDs) have thus accumulated at the foot of the deltas (Ferentinos et al., 1988; Beckers et al., 2016; Fig. 1B). Alongside these gravity-driven sedimentary processes, contour-parallel bottom-currents also influenced sediment transport in this area (Beckers et al., 2016)."

We also have clarified the section 5.2. Sediment sources in the following way: "5.2 Sediment sources

According to the mapping of the thickness of the deposits, large sliding events in the westernmost Gulf of Corinth mainly result from slope failures in, or close to, the Gilbert-type fan-deltas. Large sediment volumes were trapped in these deltas during the Holocene. As shown in Figure 1, Holocene foreset beds reach 40 to 60 m in thickness on average in the Eroneos and Meganitis fan-deltas, and sediment accumulation during the Holocene exceeding 100 m have been observed locally in between. These are the sources of MTD 10 in sliding event C and MTD 14 in sliding event D. The remarkable amount of sediments delivered to the gulf of Corinth during the Holocene probably results from large volumes of sediments stored onland during the last glacial period that were mobilized from river floodplains and colluvial deposits to rivers deltas. Widespread soil erosion resulting from human deforestation and agriculture during the second half of the Holocene also contributed to increase sediment fluxes in this period. Similarly, the previous period considered here spanning ∼130 ka to ∼11 ka is also characterized by a large sediment accummulation with a pile of 60 to 100 m forming the delta fronts of the Erineos and Meganitis delta (Fig. 1). These sources are one of the main source of MTD 10 in sliding event F."

Comments from Referee 1 : The preconditioning and triggering factors remain uncertain. I note that the point (abstract) of dramatic changes in water depth and water circulation at 10-12 ka is only applicable to a some of the cases.

Author's response: The preconditioning and triggering factors are discussed at length in the paper with section 5.4 and 5.3. Some of the context about the preconditioning factors (i.e. quantity of sediments available on slopes for mass transport), and triggering factors (fault maps, and relation between fault map and sediment accumulation on slopes) was missing and is presently displayed in Figure 1. In addition we have clarified the section 5.2. Sediment sources (see above).

Comments from Referee 1 : Landslide dynamics and the tsunami potential are briefly mentioned but not really addressed. Such assessment would re- quire modelling, but also information about the soil properties, the source areas, etc. Not all landslides will create tsunamis (see Løvholt et al., 2017).

Author's response: We fully agree that assessment about landslide dynamics and tsunami potential is not fully addressed, because it is beyond the paper scope and would require modeling. In addition, it would require a precise knowledge of the land-slide source area that we have not. We only have a first order estimate of most of the source areas (i.e. Erineos delta fan).

Comments from Referee 1 :The authors report landslide volumes, calculated from a (sparse) grid of seismic reflection profiles. The authors should mention the method used to obtain these values (e.g., gridding algo- rithm) as well as adding a statement about the uncertainty, particularly considering the line spacing of the seismic lines, and the lack of 3D seismic data. Can we be sure that the spatial extent mapped is a realistic impression of the failures or can they be over-estimated, due to the gridding and missing out areas where there are no deposits (but not evidenced because of the lack of data). This should be added as a key point under 5.1 Limitations of the analysis.

Author's response: We report landslide volumes and were extra careful in the map-ping. The gridding algorithm was specified in the text: line 100-102 "an inverse dis-tance weighted interpolation between thickness data points was used to derive isopach maps of the deposits and estimate their total volume". For the small size MTD of SED A, the comparison with other volume evaluation shows that our volume evaluation is adequate. For MTD with a large size, the volume would be adequate because of the large surface area sampled by numerous seismic profiles. We are uploading Figure 2 that show the mapping of the MTDs with the seismic grid, but we are not considering to include the seismic grid in the published version of figure 2 because it would be difficult to read it with the grid.
Author's changes in manuscript: We would include the seismic grid in the figure 7 showing the largest MTDs (MTD 10, MTD 14 and MTD 17). The figure 7 uploaded thus now shows the seismic grid and the inferred mapping that took into account the geomorphological and topographical constraints: MTD10, 14 and 19 to the north were constraints by the Trizonia scarp.

Author's response: Two versions of Figure 2 were uploaded: one as we want to include it the paper and the other with the seismic grid in response to the reviewer comment. We prefer to indicate the seismic grid in Figure 7 and not in Figure 2, because it was more difficult to read figure 2 with the grid.

Comments from Referee 1: What is the onshore-offshore relationship of the landslides?

Author's response: There is a priori no relationship. All submarine landslides originate from the submarine delta-fans. Landsliding is documented onshore on the northern coast and along the Psathopyrgos scarps, but it has no influence on the submarine landslides documented. The new figures 1 added now provide the necessary context for a better understanding to readers

Comments from Referee 1: In the interpretation, the authors repeatedly refer to blanking but they do not really illustrate what is it and what the causes may be. Author's response: We agree with the reviewer that we do not illustrate what is it and what the causes may be. So we add some more details and differentiate more clearly in the text the different blanking areas and stratigraphy that have unclear origin. First, blanking occurs below the Holocene and in two distinct spots. In the Mornos Canyon, a wide blanking area exists at a depth of about 50 to 70 m below the sea floor, a few meters below reflector 1, in direct continuity with the fan delta of the Mornos River. The origin of the blanking is unknown, but it is a low-stand related feature related to the Mornos Delta and it might correspond to coarse grained, organic rich sediments. Another area with blanking occurs at the junction between the Mornos Canyon and the

Delphic plateau at the foot of the Erineos foreset beds, at a depth similar to SE F (MTD 19); it is associated with strongly disturbed sediments forming mounds. Its origin is unknown, but it might be related to a MTD. Finally there are uncertainties regarding the southward extension of MTD 14. It extends into a zone of chaotic reflections and very disturbed seismic stratigraphy of unclear origin. Our estimate of the volume of MTD 14 was thus conservative and is considered as a minimum.

Author's changes in manuscript: To clarify the statements, we have rewritten the paragraph line 199- 207 dealing with the blanking and uncertain area in the following way. "In some zones (Fig. 2), the existence or the geometry of MTDs is difficult to evaluate because of seismic blanking and strong chaotic reflections affecting some stratigraphic intervals. Above reflector 1, the stratigraphy is clear except regarding the southern extension of MTD 14 in SE D. The low amplitude, almost transparent reflections characterizing the MTD deposit extends until a more chaotic and thicker deposit associated with surface mounds (Fig. 5). We could not decipher if the chaotic reflections that disturb the seismic stratigraphy was associated with MTD 14 in SE D or in relation with sediment remobilization from the underlying sliding event F (Fig. 4). So the mapped extension of MTD 14 in Fig. 2E is conservative and considered as a minimum. Below reflector 1, the amplitude of the reflectivity sharply decreases, which is a characteristic of lowstand deposits in the Gulf (Bell et al., 2008), and blanking occurs in two areas .In the Canyon area, a wide blanking area exists at a depth of about 50 to 70 m below the sea floor, a few meters below reflector 1, in direct continuity with the delta of the Mornos River. Blanking is thus a low-stand related feature and might correspond to coarse grained, organic rich sediments of the Mornos River. Consequently, the stratigraphy of MTDs between reflectors 2 and 1 is well established only below the Delphic Plateau. The other area associating with blanking and strongly disturbed sediments forming mounds occurs at the junction between the Canyon and the Delphic plateau at the foot of the Erineos foreset beds, at a depth similar to SE F. Its origin is unknown, but it might be related to an MTD deposit in relation with MTD 19."

Comments from Referee 1: Likewise, the authors refer to coarser grained material in a deformed mass transport deposit, but there is no evidence for this. I doubt that one would be able to observe this from sparker data, as the masses are essentially deformed. Maybe speculation?

Author's response: We evidenced that the MTD are usually lenticular bodies of low-amplitude, incoherent reflections.

Author's changes in manuscript: We removed the sentence referring to coarse-grained deposits line 140: "which would make them different from the coarse-grained deltaic deposits that are known to fail relatively frequently along the southern coast." The text is now explicit about the limit of the interpretation of the MTD facies: "In the Delphic Plateau basin (eastern part of the deep flat basin), most MTDs are imaged as lenticular bodies of low-amplitude, incoherent reflections (Fig. 3 and 4). They generally have a flat upper surface and pinch out on their margins. Their thickness ranges between a few meters, which is the minimal thickness for a MTD to be imaged with the seismic system used, and 53 meters. The geometry and seismic facies indicate subaquatic mass-flow deposits (e.g. Moernaut et al., 2011, Strasser et al., 2013). The seismic facies of many MTDs also suggests a fine-grained lithology. However, this statement must be viewed cautiously considering the uncertainties on the interpretation of seismic facies in terms of grain-size, especially for reworked sediments. For instance, failure of coarse-grained deltaic deposits commonly result to their total disaggregation and transformation into grain flows and turbidity currents, whereas finer grained deposits evolve as landslides and cohesive debris flows (Tripsanas et al., 2008)."

Author's response: But in the next paragraph (line 145-149), we still want to evidence that some MTD display a different facies with high-amplitude reflections and coherent layering, which could be related to coarse-grained sediments.

Author's changes in manuscript: We change the sentence, to evidence that it was a possible interpretation: "In the Canyon basin (western part of the deep flat basin), the MTDs present the same general characteristics but the reflector pattern is more variable (Fig. 4). Some high-amplitude reflections and coherent layering are observed in some MTDs, revealing suggesting coarser-grained sediments and locally preserved stratigraphy."

Author's response: So in the section 5.2. Sediment sources, we are discussing the observation about the two types of seismic facies observed regarding the MTD. We are also providing more information about the fact that the foresets are made of a thick accumulation of stratified fine-grained sediments and that are not made of coarse grained sediments.

Author's changes in manuscript: The text as been corrected as followed: "The seismic facies of most large MTDs also implies that they are likely composed mainly of fine-grained sediments, and seismic profiles across fan-delta area have shown that the pro-delta foresets are locally made of a thick accumulation of stratified fine-grained sediments . These fan-delta sediments are probably the main source of sediments for the largest MTDs (MTD 10, 14 and 19). However, some smaller MTDs seem to be made of coarser-grained sediments according to the seismic character (e.g., in SEs A and B in the Canyon basin), suggesting failure also occurred in coarser-grained parts of the fan-deltas located at the junction between the topset and the foreset (e.g., the 1963 slide in the Erineos fan-delta)."

Comments from Referee 1: Smaller comments: I would recommend making the seismic profiles with the same vertical exaggerations or same scales to facilitate comparison. Likewise, please add an indication on the figures where the seismic lines cross.

Author's response: We purposely chose to show the seismic profile with different vertical exaggerations, in order to be able to evidence the different features we wanted to illustrate. We purposely chose not used the same scale or vertical exaggeration for all seismic profiles because our goal is to illustrate deposits and structures that have very different sizes, from 1 to ∼10 km in length and from ∼4 to ∼40 m in thickness. If we choose the same vertical scale for all profiles, small-scale evidences will not be visible.

Comments from Referee 1: Terminology is in places confusing. I understand from this paper that landslide event actually refers to a certain interval in time (not specified) during which various landslides (with different source locations) may occur. Thus, different landslides compose a landslide event.

Author's response: Yes your understanding is correct, but because the terminology was confusing we choose to further clarify our statements.

Author's changes in manuscript: We change the related paragraphs: The stratigraphic position of MTDs in the Canyon and in the Delphic Plateau basins is not random. Most of them are clustered and are defining multi-MTDs temporal "events", based on common un-deformed underlying or overlying reflections that can be followed across the basin. Such correlations suggest that six clustered events of large submarine mass wasting occurred over the last 130 ka. Two sliding events (SE) are represented by clustered MTDs located between reflectors 2 and 1 (SE E and F). The four others occurred during the Holocene: SE D comprises MTDs deposited just on top of the reflector 1, SE C is located in the middle of the Holocene sequence, SE B somewhat higher, and finally SE A includes MTDs that outcrop at the sea floor. …. The definition of sliding events reflects a clustering of submarine landslides in a relatively short period of time. It does not necessarily imply a synchronous occurrence of all submarine landslides included in one event. Indeed, the accuracy of the correlation between separated MTDs that are interpreted to belong to the same sliding event is in the order of one or two reflections in the seismic data. Deciphering the exact MTD chronology within a sliding event was not possible because of the discontinuous character of many reflections and the relatively large distance that separates some MTDs (up to 8.5 km). This "stratigraphical" uncertainty corresponds to ∼1-2 meters of sediment so, based on sedimentation rate estimates, sliding events represent a set of MTDs that occurred over a period of 300 to 1000 years (Lykousis et al., 2007).

Comments from Referee 1 :The map should contain all geographical references used in the text. This is currently not the C2 case.

Author's response: Geographical references have been added to the new figure 1.

Comments from Referee 1 :On Figure 1, I would recommend adding a colour-coded (shaded relief or so) topographic/bathymetry map and slope map, as both are important to understand the processes. The maps should ideally cover the onshore and offshore part. Note that the "grey lines" referred to are not only the seismic grid but also bathymetric contour lines. Add the location of the Delphic Plateau, and the "Canyon".

Author's response: A new Figure 1 has been added to provide needed context taking into account remarks from the reviewer; a shaded topography was also added. The old figure 1 is now figure 2.

Author's changes in manuscript: New Figure 1 taking into account the remarks of referee 1.

Comments from Referee 1 : There are a few typos in the text - Figure 2: explain the horizons [1] and [2].

Author's response: We explain them now.

Author's changes in manuscript: In the caption of figure 2, we now state: "Figure 2. E-W Sparker seismic profile showing the mass transport deposits imaged in the Delphic Plateau basin. See the location of the profile in Fig. 1. Horizon [1] indicates the beginning of the last post-glacial transgression, at 10.5-12.5 ka and horizon [2] the marine isotopic stage 6 to 5 transgression, which occurred at ca. 130 ka (Cotterill, 2006; Beckers et al., 2015; 2016)"

Comments from Referee 1: The term "outcrop" suggests that something was eroded on top. This may not be the case for the youngest landslide deposits. Consider using exposed as the seafloor

[Figure]

Author's response: We took into account this comment.

Author's changes in manuscript: line 197-198: SE A includes MTDs that outcrop at the sea floor was change to SE A includes MTDS at or near the seafloor responsible for the present-day hummocky topography of the seafloor line 219: Sliding event A: Eight MTDs that outcrop at the sea floor have been identified. was changed to Sliding event A: Eight MTDs at or near the seafloor have been identified.

Comments from Referee 1 : Figure 6 is too small, and ideally, the maps should all use the same area, to facilitate comparison. This would be a good place to add the various source areas.

Author's response: We took into account this comment.

Author's changes in manuscript: We enlarge Figure 6 and use the same area to facilitate comparison. We also place the different source areas.

Please also note the supplement to this comment:
https://www.nat-hazards-earth-syst-sci-discuss.net/nhess-2017-371/nhess-2017-371-AC2-supplement.pdf

―――――――――――――――

**Fig. 1.** New Figure 1

**Fig. 2.** Figure 2 with grid

**Fig. 3.** New Figure 7 with grid

**Supplement:**

[revised manuscript text omitted]